# A high-throughput multiparameter screen for accelerated development and optimization of soluble genetically encoded fluorescent biosensors

Dorothy Koveal [1], Paul C. Rosen [1,2], Dylan J. Meyer [1], Carlos Manlio Díaz-García [1,4], Yongcheng Wang[3,5], Li-Heng Cai [3,6], Peter J. Chou [1,7], David A. Weitz[3] & Gary Yellen [1✉]

Genetically encoded fluorescent biosensors are powerful tools used to track chemical processes in intact biological systems. However, the development and optimization of biosensors remains a challenging and labor-intensive process, primarily due to technical limitations of methods for screening candidate biosensors. Here we describe a screening modality that combines droplet microfluidics and automated fluorescence imaging to provide an order of magnitude increase in screening throughput. Moreover, unlike current techniques that are limited to screening for a single biosensor feature at a time (e.g. brightness), our method enables evaluation of multiple features (e.g. contrast, affinity, specificity) in parallel. Because biosensor features can covary, this capability is essential for rapid optimization. We use this system to generate a high-performance biosensor for lactate that can be used to quantify intracellular lactate concentrations. This biosensor, named LiLac, constitutes a significant advance in metabolite sensing and demonstrates the power of our screening approach.

[1] Department of Neurobiology, Harvard Medical School, Boston, MA, USA. [2] Department of Biology, Massachusetts Institute of Technology, Cambridge, MA, USA. [3] Department of Physics and John A. Paulson School of Engineering and Applied Sciences, Harvard University, Cambridge, MA, USA. [4]Present address: Department of Biochemistry and Molecular Biology, University of Oklahoma Health Sciences Center, Oklahoma City, OK, USA. [5]Present address: Liangzhu Laboratory, Zhejiang University Medical Center, 1369 West Wenyi Road, Hangzhou 311121, China. [6]Present address: Department of Materials Science and Engineering, University of Virginia, Charlottesville, VA, USA. [7]Present address: Department of Biochemistry, Stanford University School of Medicine, Stanford, CA, USA. ✉email: gary_yellen@hms.harvard.edu

Genetically encoded fluorescent biosensors are important tools for studying metabolism. Their fluorescent signal provides high temporal resolution, both on fast timescales and during chronic imaging[1–5], and because they are genetically encoded, they can be directly expressed in living cells and targeted to specific cell types and organelles[6,7]. As single cells can be metabolically distinct, and metabolites are turned over at a rapid rate[8,9], biosensors have uniquely enabled precise measurements in individual cells of baseline metabolic states and metabolic perturbations in response to a challenge or stimulus[1,10].

To quantify metabolite concentrations, the readout from a fluorescent biosensor must be i) tuned to physiological concentrations of the target ligand, ii) highly specific for that ligand, and iii) robust against changes in the cellular environment (including pH) and the expression level of the biosensor itself. The only way to develop such high-performance biosensors is to screen for them, assaying many individual variants from large biosensor libraries as they are exposed to many ligand concentrations and conditions.

This poses a challenge for biosensor screening. Despite impressive advances in engineering new platforms for evolving fluorescent proteins and biosensors[7,11–14], existing screens are still limited in the number of conditions that can be tested against the biosensor. Here we have surmounted these limitations, increasing screening content and throughput by orders of magnitude, by combining droplet microfluidics and automated two-photon fluorescence lifetime imaging (2p-FLIM). Fluorescence lifetime is the average time between photon absorption and photon emission, and lifetime sensors have emerged as high-performance tools for quantifying small molecule levels in intact cells[7,15,16]. Here we have exploited microfluidically produced semipermeable gels, called gel-shell beads (GSBs)[17] that serve as microscale dialysis chambers and have used them to assay thousands of independently isolated lifetime sensor variants against many different conditions, simultaneously evaluating affinity, specificity, and response size.

We also show how our screening system, BeadScan, can be used to rapidly develop and optimize genetically encoded fluorescent biosensors. As a demonstration of our screen's capabilities, we used BeadScan to generate a high-performance lifetime sensor for lactate, named LiLac. Lactate is a key product of glycolysis that has recently been appreciated as a major cellular fuel that circulates in blood, a means of communicating redox state across cells and tissues, and a preferred fuel for certain types of cancer[18–20]. Existing lactate biosensors have been used to investigate metabolism in the brain and in cancer cells[21–26], but each has limitations that make quantitation challenging (e.g. weak sensitivity to physiological changes in lactate, or undesirable responses to calcium and/or pH). LiLac exhibits a large response size (1.2 ns lifetime change and a > 40% intensity change in mammalian cells), specificity for physiological [lactate], and resistance to calcium or changes in pH. Furthermore, the LiLac lifetime response is very precise and does not require normalization, facilitating the quantitation of lactate concentrations in living cells. Thus, LiLac constitutes a powerful tool for studying metabolism at the single-cell level and demonstrates the advantages of our screening approach.

## Results

**GSBs can be used to encapsulate, express and screen libraries of genetically encoded fluorescent biosensors**. GSBs are ideal microvessels for assaying biosensors under a series of widely varied conditions such as for a dose-response curve or specificity assay. Their semipermeable shells exchange solutes under 2 kDa[17], which allows them to retain DNA and biosensor protein while passing small molecules such as the target analytes of chemosensors. We found that GSBs stick naturally to clean glass coverslips, presumably because of adhesion between exposed positive charges in the "shell" and the negative charge on glass. By imaging GSBs containing prepurified biosensor protein, biosensor fluorescence can be measured under a series of conditions by exchanging solutions around the adherent GSBs (Fig. 1a–d). This assay format is more convenient for testing a diverse set of conditions than multiwell assay plates or microcapillaries. We can use this approach to measure biosensor responses to analytes such as ATP or NADH, showing that the polyelectrolyte shells of the GSBs permit passage even of these moderate-size biomolecules with multiple charges.

Using GSBs for a screen of many biosensor variants requires trapping a single species of DNA and its derived biosensor protein in individual GSBs, with a high enough concentration of biosensor protein to assay the fluorescence (typically in the micromolar range for fluorescent proteins). Previously, GSBs have been used for the selection of evolved enzymes by capturing and lysing individual bacteria expressing enzyme variants[17]. The enzymes are at nanomolar levels in the GSBs and their activity is visualized with fluorogenic substrates; but without the advantage of enzymatic amplification in the assay, fluorescent biosensors are difficult to detect at these low concentrations.

To surmount this problem, we developed an optimized strategy for micromolar expression of single biosensor variants in individual GSB compartments (Fig. 1e), using microbead-immobilized DNA to drive expression in an in vitro coupled transcription/translation (IVTT) system (Suppl. Fig. 1). First, individual DNA molecules from a library of variants are isolated in droplets and amplified by PCR. Next, each amplified clonal pool of DNA is captured on polystyrene microbeads via a biotin-streptavidin linkage. These clonal DNA beads are then used to drive individual IVTT reactions in droplets, which are subsequently converted to GSBs. This series of steps is accomplished by sequential use of microfluidic water-in-oil droplet formation and droplet electrofusion steps (Fig. 2) and yields GSBs with ~1000-fold higher protein expression levels than the previously published method. Conversion of a biosensor DNA library into ~$10^5$ GSBs can be done in 2 days, and ~10,000 variants can feasibly be screened in a week.

The first step in the workflow is preparing the clonal DNA library, with a sufficient number of copies to drive high expression of biosensor protein in IVTT droplets ($10^4$–$10^5$ copies/droplet). To achieve high copy numbers of individual clones, single copies of DNA are isolated in microfluidic droplets and amplified by PCR (emulsion PCR, or emPCR; Fig. 2a). Emulsifying a mixture of very dilute DNA ensures that most droplets have 1 or 0 copies of DNA. Each emPCR droplet then serves as a microscale reaction chamber for amplification of the isolated template using PCR reagents. While we did not quantify the final copy number/droplet, a 35 μm emPCR droplet can theoretically produce millions of copies of ~2 kb dsDNA before the reagents have been exhausted.

Because purified IVTT systems are highly sensitive to changes in the reaction conditions, particularly reagent dilution, directly combining emPCR droplets with similarly-sized IVTT droplets results in poor sensor expression. Therefore, we chose to immobilize the amplified DNA on beads, purify the DNA beads and then use a minimal volume to deliver them into IVTT droplets, such that the final IVTT + DNA bead droplets contain undiluted IVTT reagents with no carryover of PCR reagents.

Existing methods for producing DNA beads first bind primers to beads and then amplify from the bead surface. These methods are limited to ~$10^3$ copies/bead for large amplicons (~1–2 kb for a biosensor), presumably due to the low efficiency of amplification

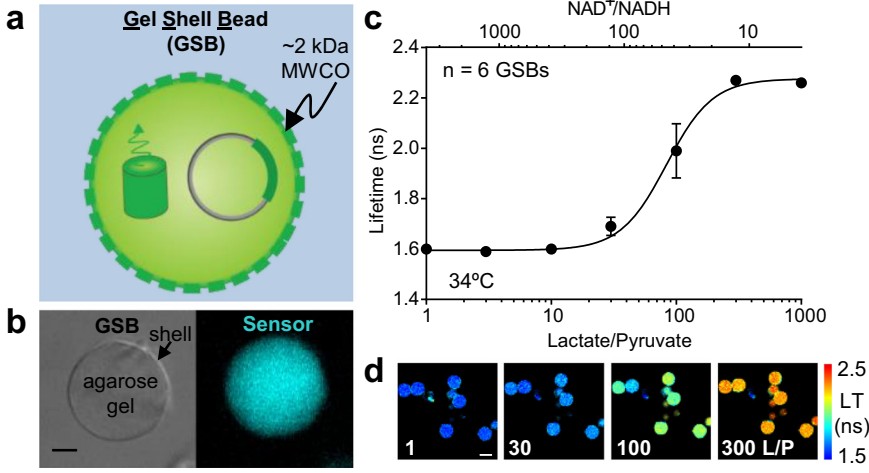

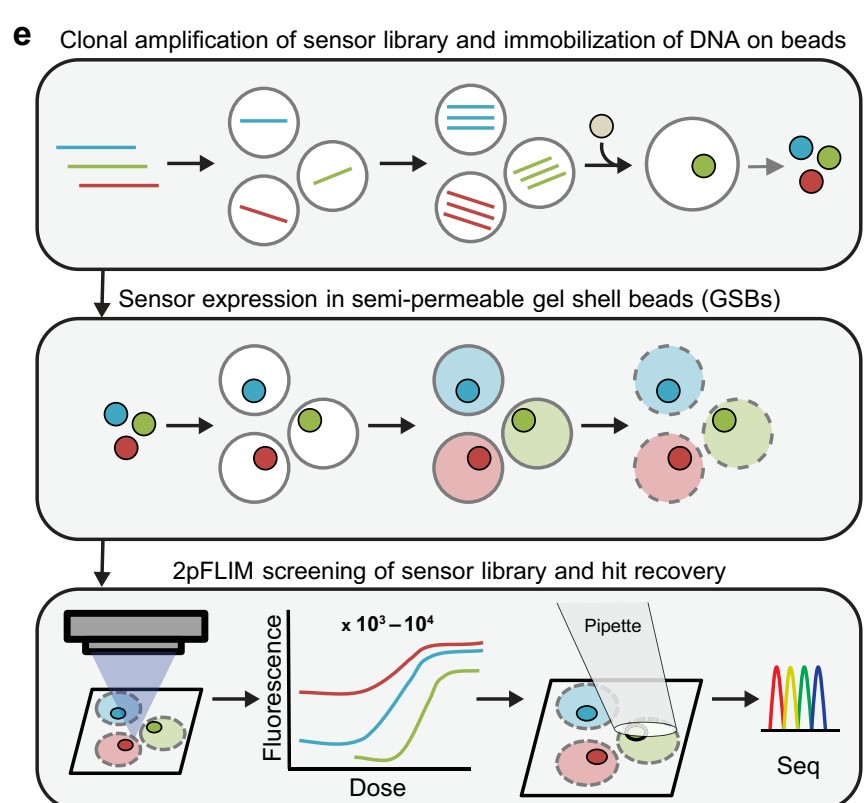

**Fig. 1 GSBs as a platform for screening genetically encoded fluorescent biosensors. a** Cartoon depiction of a GSB containing plasmid DNA and fluorescent biosensor protein. An agarose gel core is wrapped in a polyelectrolyte shell with a reported molecular weight cutoff of ~2 kDa[17]. **b** A single 20 µm GSB doped with a green fluorescent biosensor, brightfield (*left*) and epifluorescence (*right*), 5 µm scale bar. **c** Dose response curve of Peredox, a fluorescence lifetime sensor that responds to NAD$^+$/NADH[1,47], acquired at pH 7.3 at 34 °C on purified biosensor protein encapsulated in GSBs (mean ± SD). Purified lactate dehydrogenase (LDH) plus lactate and pyruvate was used to set the NAD$^+$/NADH ratio outside the GSBs, following methods previously established for Peredox[47]. LDH does not enter the GSBs. **d** Filmstrip of the fluorescence lifetime images used to generate the data in (**c**), pseudocolored according to the lifetime heatmap at the right, and arranged in order of Lactate/Pyruvate (L/P). 30 µm scale bar. **e** BeadScan overview schematic. Single plasmids from a biosensor library are isolated in microfluidic droplets that serve as picoliter-sized reaction chambers for PCR-based amplification. Clonal amplicons are immobilized on affinity beads introduced into each droplet, ultimately yielding a library of clonal DNA beads. Single beads are then re-encapsulated in new droplets containing in vitro transcription/translation (IVTT) reagents for biosensor expression. IVTT droplets containing mature biosensor are then converted into semipermeable gel-shell beads (GSBs). Up to 10,000 GSBs are arrayed on a glass coverslip for automated fluorescence imaging and analysis. Single GSBs displaying favorable biosensor properties are collected via micropipette, and the genotypes recovered from the DNA bead by PCR and Sanger sequencing.

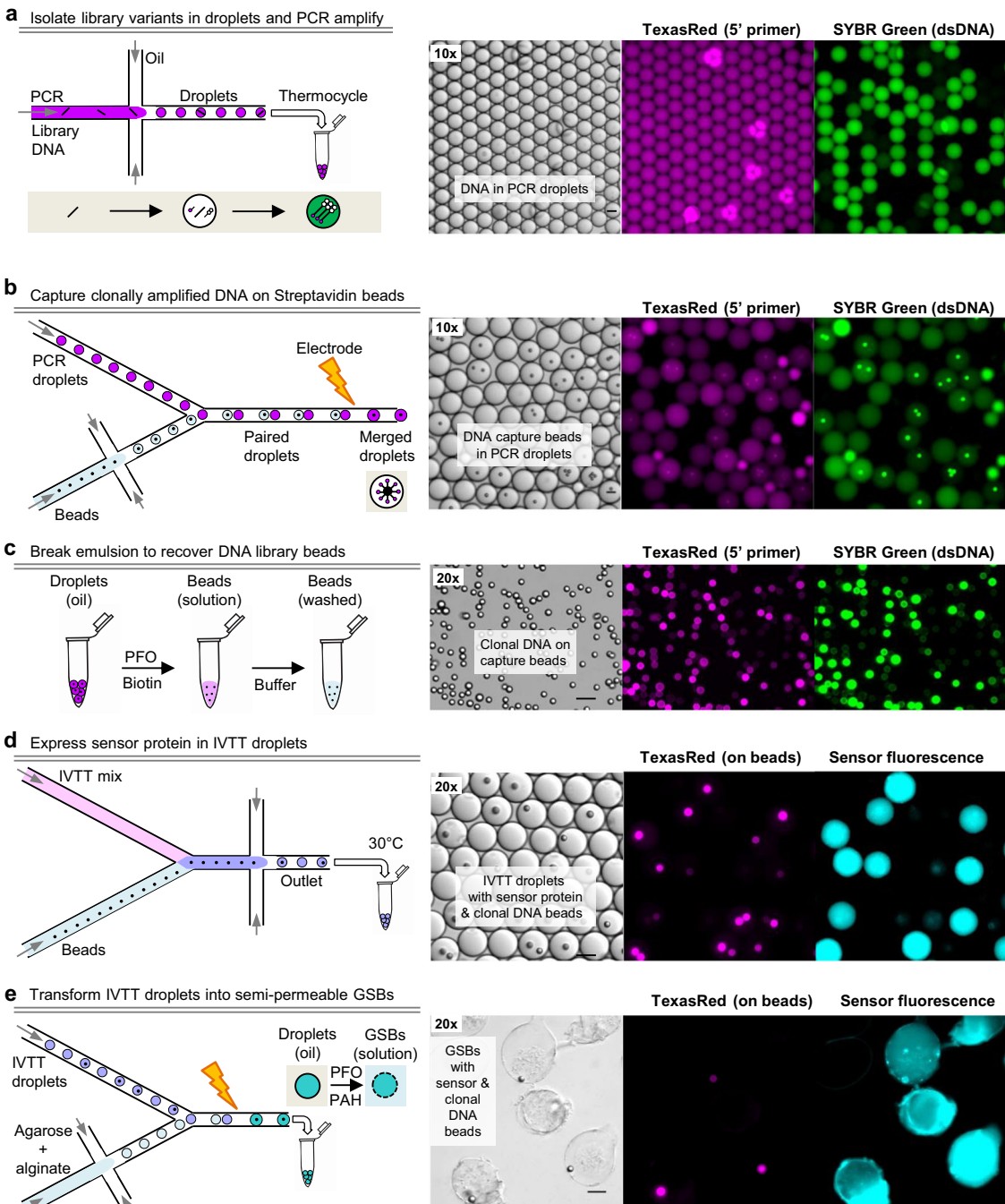

**Fig. 2 Microfluidic workflow to generate a biosensor library in GSBs. a** Dilute library DNA is mixed with hot-start PCR reagents and emulsified using a microfluidic droplet generator, such that most droplets contain either 1 or 0 copies of DNA. All droplets have 5'-TexasRed-Forward (stick with pink dot) and 5'-Biotin-Reverse (stick with open double pentagon) primers, such that amplicons carry a TexasRed tag on the sense strand and a biotin tag on the antisense strand. Droplets with amplified DNA will stain with SYBR Green, added to a diagnostic aliquot of the emulsion. **b** Biotinylated DNA is captured on streptavidin beads by controlled microfluidic merging of PCR droplets with bead-containing droplets. PCR droplets are reinjected into a microfluidic device, while a suspension of streptavidin beads is emulsified on-chip. Pairs of droplets are actively merged (>80% efficiency) as they pass a localized field generated by an electrode (10 kHz, 0.5 kV), at which point the amplified DNA is captured by the beads. **c** Addition of perfluorooctanol (PFO) breaks the emulsion, which separates into an aqueous phase containing the beads and an oil phase. The beads are then washed to eliminate residual unbound DNA. **d** The library of DNA beads is re-encapsulated in new droplets along with in vitro transcription/translation (IVTT) reagents using a custom microfluidic device that combines the two aqueous streams immediately prior to emulsification, mitigating premature transcription and mRNA cross-contamination. Only droplets containing a DNA bead will express fluorescent biosensor protein following an overnight incubation at 30 °C. **e** Finally, IVTT droplets are converted into durable semipermeable GSBs by controlled microfluidic merging with a mixture of agarose (gel in sol) and alginate (polyanion), followed by emulsion breakage by PFO in the presence of PAH (polycation). The two polyelectrolytes form a semipermeable shell at the surface of the gel scaffold, trapping proteins and DNA inside the gel but allowing small molecules (<2 kDa) to pass through[17]. Continuous exchange of ligands allows for the collection of full dose-response curves from the biosensor protein encapsulated in each gel during downstream screening. Scale bars represent 20 μm.

from a solid surface[27,28]. In our hands, these methods failed to reliably produce beads loaded with high copy numbers of full-length amplicons; only 0.1–1% of the resulting DNA beads were capable of driving strong expression of our target sensors in IVTT droplets. To overcome these limitations, we instead allowed template DNA to be amplified free in solution in droplets containing a biotinylated 3' primer, and then subsequently immobilized the PCR products on streptavidin beads. To achieve this, each amplified emPCR droplet is fused with a paired droplet containing a streptavidin affinity bead (Fig. 2b) via controlled active microfluidic merging at a rate of ~4–5 million droplets per hour. The streptavidin bead captures the amplified biotinylated DNA, such that each bead is coated in many copies of a single clone. The beads are then released from the droplets and excess DNA is washed away (Fig. 2c). Using this method, a 6 μm polystyrene bead can capture >200,000 clonal copies of >2 kb amplicons, and millions of beads can be prepared in parallel. However, optimal expression of soluble sensor protein was achieved with beads intentionally limited to ~100,000 copies of DNA by pre-blocking a subset of streptavidin binding sites, because more densely loaded beads sometimes led to the accumulation of visible protein aggregates within the droplet (Suppl. Fig. 1c). We also optimized the concentration of biotinylated 3' primer included in the emPCR droplets. We found that using an excess of biotinylated 3' primer led to poor results, presumably because the unextended original biotinylated 3' primer competes with fully-extended biotinylated DNA for streptavidin binding sites. Therefore, we used a limiting concentration of the 3' primer to ensure the complete extension of all biotinylated primer molecules, so that only fully extended amplicons are captured on the streptavidin beads.

To express the biosensor library, single DNA beads are purified and re-encapsulated in droplets containing IVTT reagents using a two-stream co-flow droplet generator device that introduces beads into the IVTT stream immediately prior to droplet encapsulation (Fig. 2d, Suppl. Fig. 2a–b). After testing multiple cell-free expression systems, including cell lysates, we achieved optimal biosensor expression levels with purified IVTT reagents, specifically the PUREfrex2.0 system (Suppl. Fig. 1).

Once the biosensor protein has been expressed, IVTT droplets are transformed into GSBs by: 1) merging single IVTT droplets with droplets containing a mixture of agarose (gel) and alginate (polyanion), 2) dispersing the droplets in a polycation emulsion (poly(allylamine)hydrochloride, PAH; Fig. 2e, Suppl. Fig. 2c–d), then 3) breaking the mixed emulsion to allow complexation of the two oppositely charged polymers. A semipermeable shell forms at the surface of the agarose gel matrix, trapping the biosensor protein inside but allowing small molecules to pass through[17]. In initial tests, we observed a ~50–60% loss of sensor protein during the final step. While the rate of shell formation is undetermined, we presume that upon emulsion disruption, some protein diffuses out of the gel before the shell can form. Therefore, we included Ni-NTA nanospheres in the gels to retain the his$_6$-tagged sensor protein during shell deposition. Once the shell has formed around the mature GSB, sensor protein is released from the nanospheres with mild EDTA treatment. This modification improved sensor protein retention from IVTT droplets to mature GSBs to ~90–100%.

In its final form, each GSB serves as a microscale dialysis chamber containing a unique biosensor variant. The immobilized GSBs (~10$^5$ GSBs on a 1 cm$^2$ glass coverslip) can withstand >20 mL/min flow rates within a standard microscopy perfusion chamber, allowing rapid exchange of external conditions while fluorescence responses from the encapsulated biosensors are recorded using 2p-FLIM. After screening, single GSBs are retrieved with a microcapillary and DNA sequences

are recovered by PCR amplification and Sanger sequencing (Suppl. Fig. 3).

Because this system utilizes gel-shell beads to rapidly scan biosensor libraries for optimized properties, we have named it BeadScan. While BeadScan can be used to track changes in either fluorescence intensity or lifetime, we find that lifetime measurements are more robust against the cumulative effects of photobleaching and yield precise dose responses from single GSBs.

**Development of a high-performance lifetime sensor for lactate using BeadScan.** To demonstrate the capabilities of our system, we set out to generate a de novo genetically encoded biosensor for lactate. An existing FRET biosensor for lactate, Laconic, has been used to investigate neuronal metabolism[1,21,29,30], but its shallow response to lactate and optical complications arising from using two fluorophores make quantitative use challenging. Newer single fluorophore biosensors, GEM-IL, Green Lindoblum, CanlonicSF and eLACCO1.1 also have limitations[23–26]. GEM-IL responds to physiological pH changes, complicating accurate lactate measurements; Green Lindoblum has a very high affinity for lactate outside of the physiological regime; and CanlonicSF and eLACCO1.1 respond to both lactate and calcium, restricting their utility to compartments with saturating levels of calcium. CanlonicSF and eLACCO1.1 are based on the TTHA0766 periplasmic lactate- and calcium-binding protein from *T. thermophilus*, while all other existing lactate biosensors use the LldR transcription factor from *E. coli* and *C. glutamicum*.

Given the limited success in improving LldR-based biosensors and the calcium sensitivity of TTHA0766-based biosensors, we selected an untested scaffold for our biosensor: the extracellular dCACHE domain of the bacterial chemotaxis protein, TlpC from *H. pylori*. TlpC is highly specific for lactate, showing no detectable affinity for structurally similar ligands like pyruvate or oxaloacetate[31]. While only the ligand-bound crystal structure of the TlpC lactate binding domain has been solved, we reasoned that lactate binding may generate a large conformational change between the N- and C- termini, which are in close proximity[31,32]. Indeed, the structurally-similar chemotaxis receptor, TlpA, can accommodate large structural shifts up to 8 Å between the N-terminal stalk helix and the C-terminal β-strand that connect to transmembrane domains[33]. To leverage this motion, the N- and C- termini of TlpC were connected to the green fluorescent protein T-Sapphire (Fig. 3a), which has yielded a fluorescence lifetime change in several previous single-color biosensors[1,10,15].

Because the linkers connecting the fluorescent protein and the scaffold can profoundly affect the biosensor response, we generated a library of TlpC-TS biosensor variants that sampled 49,152 different linkers, varying both length and composition (Fig. 3a), and screened this library using BeadScan (Fig. 3b–c). We isolated two high-contrast biosensors, TlpC-TS #1537 and #1059, displaying an inverted lifetime change ($|\Delta LT| = 0.2$ ns and 0.3 ns, respectively) with apparent affinities ($K_{app} = 0.19 \pm 0.13$ mM and $0.66 \pm 0.11$ mM, respectively; Fig. 3d–e) within the physiological range of lactate, which circulates at 1–2 mM[18,20,34]. Intracellular lactate concentrations vary widely across cell types and physiological states[35,36], but are typically expected to be within 0.5–5 mM.

**Multiparameter screening yields a biosensor that can be used to quantitatively measure lactate concentrations in living cells and tissue.** Upon further in vitro characterization, we found that TlpC-TS was highly specific for its cognate ligand over other chemically similar ligands, but was also sensitive to pH changes within physiologically relevant ranges (Fig. 3f). Many single fluorophore biosensors respond to pH, including all of the

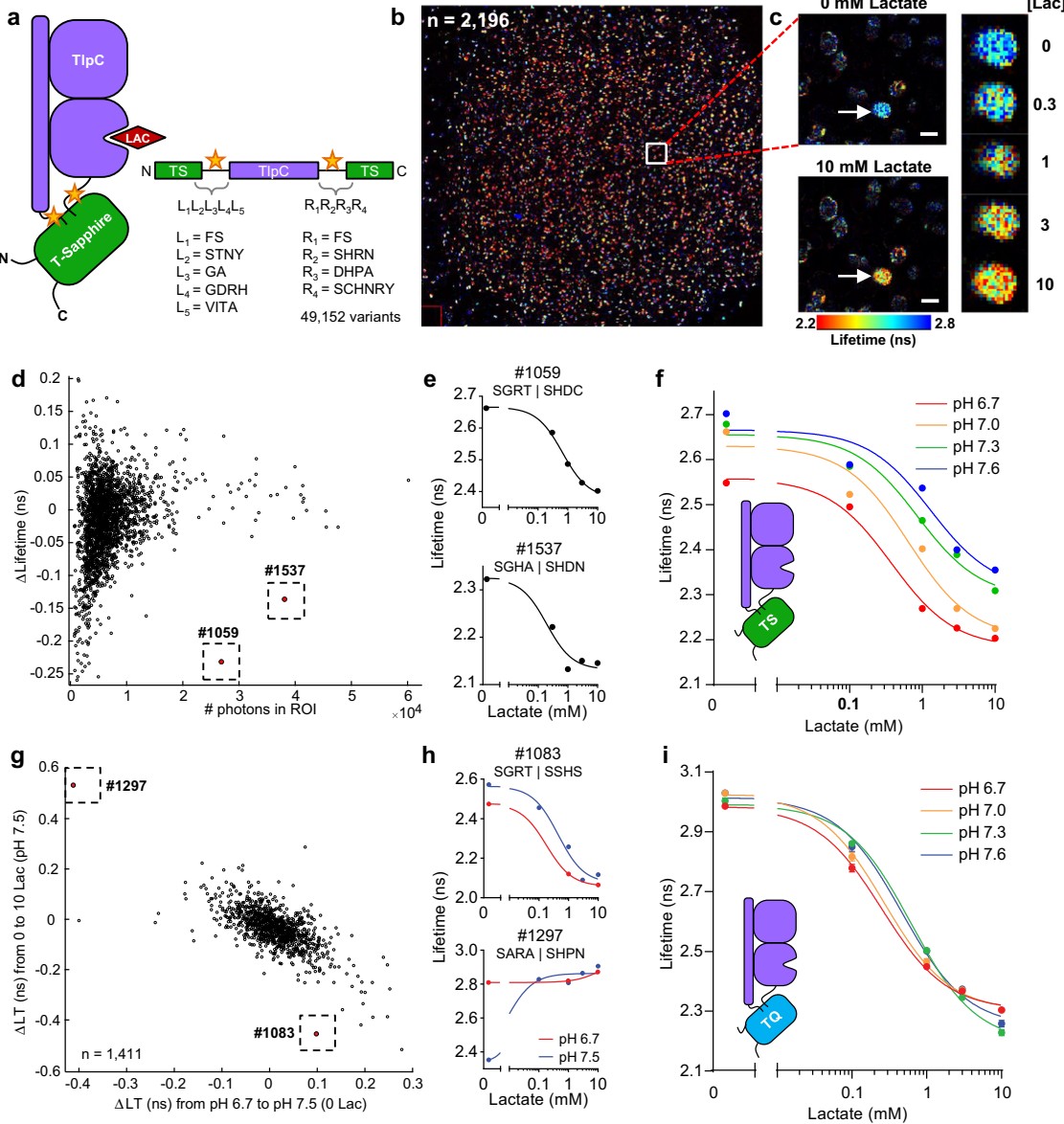

**Fig. 3 Development and optimization of a high-performance lifetime sensor for lactate using BeadScan. a** Schematic of the lactate biosensor design. The lactate-binding domain of the TlpC protein was inserted into the fluorescent protein, T-Sapphire. The flexible linker regions (yellow stars) connecting the two domains were varied in a large biosensor library, sampling 49,152 biosensor variants. **b** ~10,000 GSBs containing the TlpC-TS lactate biosensor library were immobilized on a 1-cm² glass coverslip inside a perfusion chamber. 2196 GSBs met the minimal fluorescence threshold for ROI detection. The stitched image is recreated from 121 individual frames recorded using 2p-FLIM. The white box marks a single frame (771 µm square). **c** One frame showing a single GSB (arrow) containing a biosensor variant that responded well to lactate. The biosensor displayed high fluorescence lifetime at low [lactate] and low lifetime at high [lactate]. Lifetime values are pseudocolored according to the heatmap at the bottom. Scale bars indicate 100 µm. **d** Initial screen of the TlpC-TS lactate biosensor library, showing data from individual biosensor variants captured in GSBs. Average photon counts within each ROI are plotted against the maximum change in lifetime between 0 and 10 mM lactate; each data point represents a single GSB. Two GSBs with large lifetime changes were selected (dotted boxes). **e** Dose-response curves from each of the two individual GSBs denoted in (**d**). **f** Performance of TlpC-TS #1059 at four different pH values. Dose-response curves were collected on a uniform sample of GSBs expressing TlpC-TS #1059 ($n = 96$ GSBs, mean ± SD). **g** Multiparameter screening of a linker library of TlpC-TQ, comparing lifetime changes in response to lactate versus pH for each of 1411 individual GSBs. Despite a general correlation between lactate- and pH-induced responses, rare variants that resisted this trend could be isolated (#1083). **h** Lactate dose-responses from each of the two individual GSBs denoted in (**g**) at pH 6.7 and pH 7.5. The pH-resistant variant (#1083, top) was renamed LiLac. **i** LiLac responses to lactate are highly resistant to changes in physiological pH. Data collected at room temperature using a uniform sample of GSBs expressing LiLac ($n = 90$ GSBs, mean ± SD). In (**f**, **i**), some error bars are occluded by the data points.

aforementioned lactate biosensors, except CanIonicSF (which responds to calcium). Because pH is not constant within living cells[37], data from pH-responsive biosensors requires careful interpretation; and while it is popular to calibrate pH responses with a null biosensor, the occupied and unoccupied biosensor often display different sensitivities[38–45], making this an imperfect adjustment.

We therefore set out to reduce the pH responses of our prototype lactate biosensor to enable more accurate quantitation of intracellular lactate concentrations. We reasoned that the pH

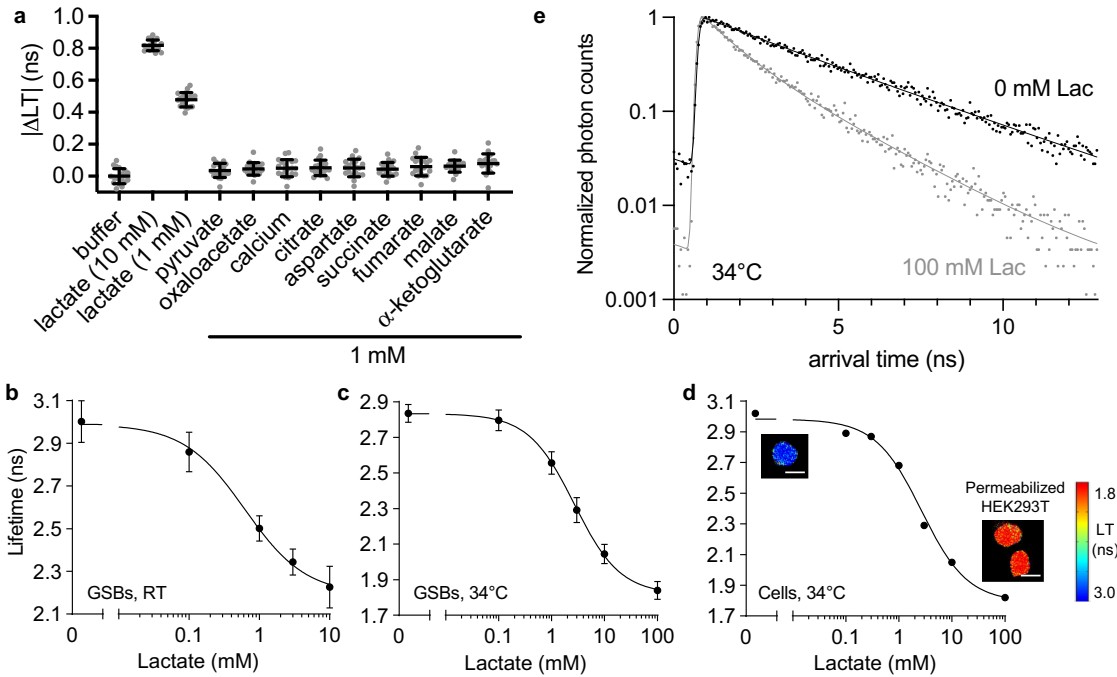

**Fig. 4 LiLac is a sensitive, specific biosensor that reports on intracellular lactate concentrations. a** Specificity testing shows that LiLac displays a change in lifetime only when exposed to lactate. A uniform sample of GSBs expressing LiLac was exposed to 10 mM or 1 mM lactate, or 1 mM of each of nine other chemical compounds that may interfere with lactate biosensors ($n = 19$ GSBs, error bars are mean ± SD). | $\Delta$LT | (ns) indicates the magnitude of the lifetime change relative to buffer. **b** A uniform sample of GSBs expressing LiLac was exposed to different lactate concentrations at pH 7.3 at room temperature (RT, $n = 90$ GSBs) or at **c** 34 ± 1 °C ($n = 21$ GSBs, mean ± SD plotted). **d** Biosensor calibration performed in permeabilized HEK293T cells at 34 ± 1 °C. Cells were permeabilized with β-escin in the presence of the lactate dehydrogenase inhibitor GSK-2837808A (2 μM), then imaged after exposure to different concentrations of lactate (4–12 cells per data point, mean ± SD). Insets of lifetime images of cells in 0 and 100 mM lactate depict fully permeabilized (swollen) cells. Pseudocoloring reflects empirical lifetime values according to the heatmap. Scale bars represent 20 μm. **e** Time-resolved single-photon arrival histograms for LiLac expressed in permeabilized HEK293T cells from (**d**), in the presence (gray dots) and absence (black dots) of lactate, each based on an image of a single cell averaged over 10 frames. Fit parameters for 0 mM lactate were $A_1 = 0.31$, $\tau_1 = 2.47$ ns, $A_2 = 0.69$, $\tau_2 = 3.81$ ns, Gaussian $\sigma = 0.078$ ns and $\chi^2 = 1.09$; and for 100 mM lactate they were $A_1 = 0.56$, $\tau_1 = 0.93$ ns, $A_2 = 0.44$, $\tau_2 = 2.3$ ns, Gaussian $\sigma = 0.078$ ns and $\chi^2 = 1.11$.

response may arise from the T-Sapphire fluorescent protein[46], and we repeated our linker library using mTurquoise2, a highly pH-resistant fluorophore recently used to develop a fluorescent lifetime sensor for calcium levels[7], which is also brighter and more photostable than T-Sapphire[7,14,46]. Using BeadScan, we screened the new library (referred to herein as TlpC-TQ) for improved lactate responses with reduced pH sensitivity (Fig. 3g–h).

For the majority of TlpC-TQ biosensor variants, the magnitude of the lactate response was correlated with that of the pH response (Fig. 3g). One variant in particular, TlpC-TQ #1297, had the same response to a change in pH (7.5 to 6.7) as it did to the addition of 0.1 mM lactate (Fig. 3H). Nevertheless, out of 1,411 variants we successfully isolated a rare variant, TlpC-TQ #1083, that displayed large lactate responses and greatly diminished pH sensitivity (Fig. 3h–i; Suppl. Fig. 4). It also exhibited a larger dynamic range with high sensitivity to physiological levels of lactate ($K_{app} = 0.62 \pm 0.04$ mM at pH 7.3 at 24 °C), features that were co-selected using our screening modality. We named this optimized lifetime lactate biosensor LiLac (DNA and protein sequences in Suppl. Note 1).

We characterized the photophysical parameters (Suppl. Fig. 5, Suppl. Table 1) and the performance of LiLac with respect to specificity and temperature in several contexts, beginning with a uniform sample of IVTT-produced LiLac in GSBs. LiLac is highly specific for lactate over other chemically similar ligands, including pyruvate, as well as calcium (Fig. 4a). The total lifetime excursion of ~0.8 ns is similar to other high-performance lifetime sensors including Tq-Ca-FLITS and Peredox (~1.3 ns and ~0.8 ns,

respectively; Fig. 4b)[7,47]. At higher temperatures, the dynamic range increases (~1 ns at 34 °C) while biosensor affinity for lactate decreases ($K_{app} = 2.68 \pm 0.15$ mM at pH 7.3 at 34 °C; Fig. 4c).

We extended our characterization of LiLac to mammalian cells, as biosensor properties can deviate slightly between in vitro and in-cell calibrations[10,40,48]. LiLac was bright and well-expressed in the cytosol of mammalian cells. Permeabilized HEK293T cells expressing LiLac displayed lifetime values between 3.0 and 1.8 ns in response to changes in perfused lactate ranging from 0–100 mM, with a $K_{app}$ of $2.66 \pm 0.18$ mM measured at 34 °C (Fig. 4d–e).

The large response and high brightness of the LiLac biosensor led to very robust signals in mammalian cells, with the lifetime measurements from individual cells subjected to changes in lactate being superimposable (Fig. 5a–b). When compared with the calibration curve obtained from permeabilized cells, intact cells displayed slightly higher lifetimes, suggesting lower lactate levels inside cells relative to the external condition, and possibly reflecting differences in lactate entry through the transporter as external [lactate] increases[35,36,49].

We compared LiLac's performance to that of the existing lifetime-compatible lactate biosensor Laconic. While the Laconic output is generally interpreted as a FRET ratio, the donor species also produces a measurable change in lifetime, as is typical of FRET biosensors. In cells, LiLac displays a ~6-fold larger response size, >5-fold reduced variability, and substantially higher signal to noise ratio (LiLac$_{SNR}$ ~ 25, Laconic$_{donor\ lifetime,SNR}$ < 1; Suppl. Fig. 6a–c). Here, the SNR metric estimates the change in lifetime signal in cells from 0 to 10 mM external lactate relative to the

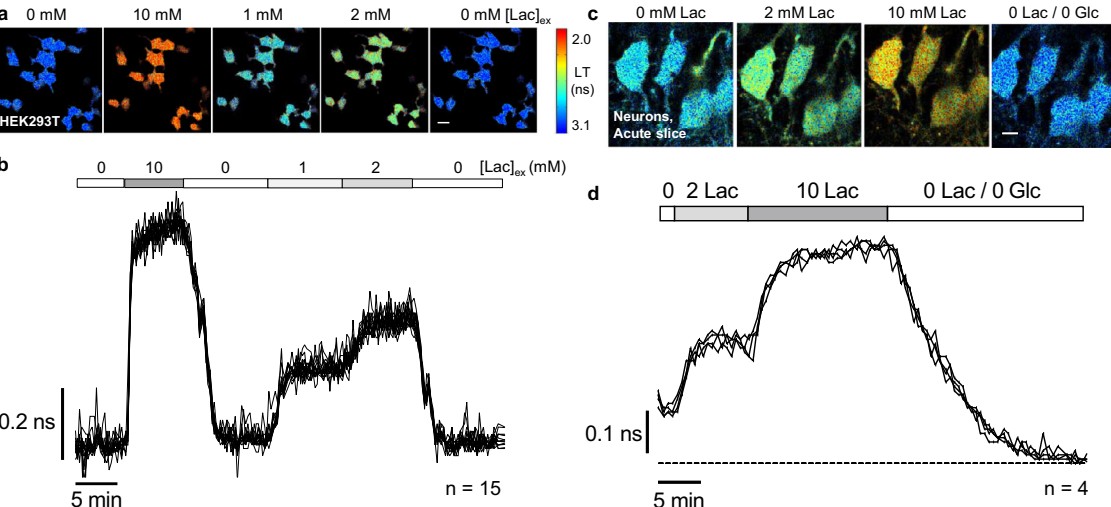

**Fig. 5 LiLac reports on changes in intracellular lactate concentration in cultured mammalian cells and acute brain slice. a** Filmstrip of lifetime images showing intact HEK293T cells expressing LiLac at 34 ± 1 °C. Lifetime values are highly uniform across a population of cells exposed to a single concentration of lactate and are responsive to changes in lactate concentration in the external bath solution ([Lac]$_{ex}$). Cells were incubated in 0 lactate/0 glucose prior to imaging. 20 μm scale bar. **b** Overlaid lifetime traces from each of 15 cells observed in (A) show that LiLac yields highly reproducible measurements of intracellular lactate concentrations across a population of cells and is sensitive to even modest manipulations of extracellular lactate (1 to 2 mM). Measurements recorded every 15 s. **c** Filmstrip and **d** single-cell traces from hippocampal neurons in acute slice expressing LiLac. Slices in 10 mM glucose were exposed to 0, 2 or 10 mM lactate, after which both lactate and glucose were washed out. Measurements recorded every 30 s. 15 μm scale bar. Images are pseudocolored by empirical lifetime values according to the heatmap, and traces indicate average lifetimes calculated over an 8 ns arrival time window ($\tau_8$).

average noise level at each condition. We also observed that the lifetime of the Laconic donor species varies widely across cells bathed in lactate (Suppl. Fig. 6b–c), as also observed with its FRET signal[23]. By comparison, the high precision of the LiLac readout suggests that the cell-to-cell variability observed with Laconic may be an artifact of the biosensor. While the source of this variability is undetermined, recent evidence suggests that Laconic may be partially degraded in cells, producing some fluorescence signal that is uncoupled from lactate binding[23].

Building on LiLac's strong performance in cultured cells, we next examined its performance in acute hippocampal brain slices (Fig. 5c–d). LiLac was well-expressed in neurons and highly responsive to bath application of different concentrations of lactate, exhibiting a tight distribution of lifetime values across neurons, as was observed of its performance in cultured cells. Lifetime values in the final condition of 0 mM lactate/0 mM glucose were ~0.1 ns higher than in the starting condition of 0 mM lactate/10 mM glucose, demonstrating that LiLac can report on perturbations in baseline [lactate] in ex vivo preparations.

Collectively, these data demonstrate that LiLac is a high-performance fluorescence lifetime sensor for lactate, and that changes in LiLac lifetime can be interpreted quantitatively. LiLac is also a high-performing lactate biosensor when used without lifetime measurement, simply by monitoring changes in intensity. Exposing mammalian cells to 10 mM lactate produces a sizeable intensity change of >40% (Suppl. Fig. 7), which is comparable to that of the most sensitive intensity-based lactate biosensor[23], but with the added benefit of substantial pH resistance. Therefore, LiLac can also be used as an intensiometric biosensor, although additional normalization would be required for the quantitative determination of lactate levels.

## Discussion
Here we have described a screening modality, BeadScan, that leverages droplet microfluidics and automated fluorescence

imaging to rapidly assay large libraries of genetically encoded fluorescent biosensors and isolate biosensors optimized on multiple features. Using BeadScan, we engineered a high-performance fluorescent lifetime sensor for lactate, named LiLac. The scale and richness of the fluorescence measurements enabled by our system accelerate screening throughput by an order of magnitude, which allowed us to more efficiently and more thoroughly assay biosensor performance.

To isolate high-performance biosensors, large biosensor libraries must be tested against many different conditions. Existing screening methodologies are limited in the number of conditions that can be assayed, constraining which performance parameters can be feasibly tested. BeadScan surmounts these limitations and also increases screening throughput. Screening approaches of comparable throughput (e.g. FACS and SortSeq)[50–52] report on a limited set of parameters (e.g. biosensor brightness) tested under a single manipulation (e.g. the presence or absence of ligand) and are not compatible with fluorescence lifetime measurements. Even sophisticated lifetime-compatible screens are still limited in the number of conditions that can be tested[13]. The semipermeable microgels (GSBs) utilized in our screening system allow the complete exchange of conditions, such that the encapsulated biosensor variants can be assayed against multiple conditions. This allows for the collection of full dose-response curves on large libraries, providing information on ligand-binding affinity, specificity and response size.

To adapt GSBs for screening fluorescent biosensors, it was necessary to increase protein expression levels by ~1,000-fold. We achieved this by devising a microfluidic pipeline that begins with the immobilization of a PCR-amplified DNA library on affinity beads, followed by the expression of individual biosensor variants in microfluidic droplets, which are finally transformed into GSBs. Our approach to "bead PCR" improves upon previously reported methods, yielding >200,000 clonal copies of >2 kb dsDNA per 6 μm bead. Existing methods that immobilize primers on a bead and then amplify from the bead surface (e.g. BEAMing) are

limited to ~1000 copies/bead for comparably sized amplicons and beads[27,28]. We presume that molecular crowding constraints at the bead surface limit amplification efficiency, while our method circumvents this problem by allowing the template DNA to be amplified in free solution in the droplet before immobilization on beads. This advance combined with optimization of the reaction conditions within the droplets allowed us to achieve micromolar expression levels of unique biosensor protein variants in IVTT droplets and subsequent GSBs.

Our high-throughput multiparameter screening approach accelerated the development and optimization of LiLac, a high-performance lifetime sensor for lactate. The initial variant of our TlpC-based lactate biosensor displayed undesirable responses to changes in physiological pH. We reduced these responses by exchanging the T-Sapphire fluorophore for mTurquoise2, a lifetime-responsive pH-resistant cyan fluorescent protein, and re-screening for variants that displayed both large lactate responses and minimal pH responses. This manipulation required linker re-optimization, which is labor-intensive using other screening methodologies, but routine using BeadScan. From a single screen, we were able to isolate from over one thousand variants a single biosensor that displayed 1) reduced sensitivity to [H$^+$] (or pH), 2) improved dynamic range, and 3) an affinity for physiological levels of lactate. We named this highly optimized biosensor LiLac.

In mammalian cells, the lifetime response from LiLac outperforms that of Laconic, the first and most widely used genetically encoded lactate biosensor, exhibiting a 6-fold larger response size and high sensitivity (SNR ~25). LiLac is also well-expressed and responsive in human cells and acute brain slice, demonstrating that it can be used to report on intracellular lactate concentrations in cultured cells and ex vivo preparations. Indeed, calibration of the LiLac response in permeabilized HEK293T cells was remarkably similar to that in IVTT-produced GSBs, showing a moderate improvement in sensor excursion with no significant change in $K_{app}$. The TlpC extracellular domain used in the LiLac scaffold does have a second putative binding domain (the distal PAS domain) that may recognize an as-yet-unidentified ligand[31] and could have an unknown effect on sensor output. However, the similarity of LiLac's performance in vitro and in cells suggests that the LiLac response is indeed highly specific for lactate in the context of the cellular environment. Most importantly, LiLac is fine-tuned for quantitative measurements, as the lifetime signal does not require pH calibration or normalization for motion artifacts or protein expression levels.

While the LiLac lifetime response is highly reproducible and easy to quantify, its intensity response can also be used to monitor lactate levels. Fluorescence intensity measurements are widely accessible, and LiLac exhibits a sizeable intensity change comparable to other intensity-based sensors, but with the added benefit of substantial pH resistance. As demonstrated here, LiLac intensity ($\Delta F/F$) can be used to track intracellular lactate changes in single cells over time. Quantitative interpretation of the intensity signal would require additional adjustment for variable expression levels and motion artifacts, often achieved by normalization to a covalently-attached inert red fluorophore, such as mCherry. However, normalization accuracy would be subject to the many well-documented caveats associated with two-color imaging[38,53].

The rapid isolation and high performance of LiLac demonstrates the advantage of our screening approach, which significantly improves sensor screening throughput and content, and enables co-evaluation of multiple sensor parameters (response size, affinity, specificity/pH response, intensity) for entire libraries. Even though LiLac was screened in an in vitro environment, it translated to cells remarkably well. We believe that LiLac's success

arises from how thoroughly we were able to optimize its lactate response, tuning its affinity to match the physiological range and making it robust against other potential ligands (most notably, H$^+$) present in the cellular milieu. Ions, including H$^+$, are important ligands. Indeed, the structurally-similar TlpB receptor directs pH-taxis via a pH sensing mechanism in the distal PAS domain of its extracellular binding domain[33]. While the TlpC libraries were not screened against other small molecules, like pyruvate or oxaloacetate, as these parameters had previously been established for the TlpC binding domain[31], BeadScan can support specificity screens against co-varied small-molecule ligands. We anticipate that other soluble sensors subjected to similarly stringent screening parameters would also translate well into cells.

We also envision that other labs will be able to implement the BeadScan system. Microfluidic start-up costs are on par with that of 96-well plate screening (four of our microfluidic setups can be built for the cost of one plate reader) and the microfluidic devices described here are commercially available. The cost of one microfluidic droplet generator device is comparable to that of a miniprep, and because of the low volume requirements the reagent costs are negligible. Of course, the equipment required for automated fluorescence imaging is the largest contributor to start-up costs. Here, we have used an automated 2p-FLIM microscope to collect fluorescence lifetime responses of the sensor libraries. However, BeadScan could readily be adapted for fluorescence intensity-based screening on an epifluorescence microscope equipped with a motorized stage. Relative to 96-well plate screening, BeadScan enables significantly higher throughput, saving time and further reducing run costs. A typical GSB sample can be prepared in two days, yielding up to five coverslips for screening, each with up to ~2,000 variants isolated in GSBs. It is therefore quite practical to screen tens of thousands of isolates each week. While we isolated LiLac by screening just one coverslip, other screens seeking very rare variants could greatly benefit from this higher level of throughput. And while the throughput alone is remarkable, the most notable breakthrough stems from the ability to assay multiple conditions against single isolates. This capability enables co-optimization for complex sensor features, including affinity, contrast, and specificity/pH sensitivity, setting BeadScan apart as a screening methodology.

BeadScan does have some limitations. We are currently unable to evaluate biosensor kinetics within the GSB environment, which would require a precise understanding of ligand diffusion rates across the polyelectrolyte shell. BeadScan is also most appropriate for screening soluble biosensors, not membrane-associated or GPCR-based biosensors[54,55]. Conversely, BeadScan does have the potential to screen for other biosensor parameters not used here, such as ligand specificity, photostability or photochromism. And while we have demonstrated that BeadScan yields precise dose responses based on fluorescence lifetime, it can also be used to measure fluorescence intensity responses with careful correction for protein expression levels, motion artifacts, and photobleaching effects.

Future improvements to the system may address ways to capture kinetic information on sensors within the GSB environment, or the efficiency of GSB generation. Because the two droplet generation steps randomly sort particles (e.g. single pieces of DNA or individual DNA beads) into droplets, not every GSB encodes a sensor. Random sorting is inefficient, and other systems that use large deformable hydrogel beads for DNA delivery are able to beat Poisson statistics, synchronizing beads to achieve near-perfect delivery into droplets (e.g. ~100% of droplets contain exactly one hydrogel bead)[56,57]. While the small polystyrene beads used in our system are not deformable, it may be possible to adjust the system to enable synchronized bead delivery. Such a

modification could significantly improve screening efficiency and throughput.

In conclusion, we have presented a screen for highly efficient multiparameter optimization of fluorescent biosensors, and we have used it to generate a lactate biosensor, LiLac, that can be used for quantitative evaluation of lactate concentrations in living cells. We anticipate that this screening approach will be generalizable to other soluble biosensors, and that LiLac will enable valuable new experiments probing lactate metabolism.

## Methods

**Ethical statement**. Our study complies with all relevant ethical regulations established by the Harvard Committee on Microbiological Safety, the NIH Guide for the Care and Use of Laboratory Animals and Animal Welfare Act, and the Harvard Medical Area Standing Committee on Animals (protocol #IS00001113, assurance A3431-01).

**Statistics and reproducibility statement**. No statistical method was used to predetermine sample size. No data were excluded from the analyses. The experiments were not randomized. The Investigators were not blinded to allocation during experiments and outcome assessment.

**Microfluidic device operation**. Microfluidic devices were purchased from Droplet Genomics (Vilnius, Lithuania). Custom microfluidic device schematics are presented in Suppl. Fig. 2. Flow rates of the liquid and oil (QX200™ Droplet Generation Oil for EvaGreen, BioRad) phases were controlled using custom in-house 3D printed syringe pumps operated by an Arduino UNO connected to two X-NUCLEO-IHM02A1 two-axis stepper motor driver expansion boards. Each of the phases was injected into PDMS devices via medical-grade polyethylene microtubing (I.D. × O.D.: 0.015″ × 0.043″ / 0.38 mm×1.09 mm, Scientific Commodities Inc, BB31695-PE/2). Images were captured using a high-speed camera (Pixelink, PL-D732MU-NIR-T) mounted on the eyepiece port of a Zeiss IM35 inverted microscope.

**Fluorescence microscopy**. Droplets were imaged on a Nikon TiE inverted microscope equipped with an Andor Revolution DSD spinning disk unit using a Plan Fluor 10X/0.3 N.A. objective or a Plan Apo VC 20X/0.75 N.A. objective (Nikon) illuminated with an LED light source (Spectra X; Lumencor, Beaverton, OR). Excitation light was passed through a bandpass filter (578/16 nm for TexasRed, 482/18 nm for SYBR Green, or 445/20 nm for T-Sapphire-based biosensors). Red emission was collected through a 629/56 nm bandpass filter, following a 590 nm shortpass dichroic. Green emission was collected through a 525/39 nm band pass filter, following a 490 nm short pass dichroic. Images were analyzed using ImageJ 1.53c.

**Two-photon fluorescence lifetime imaging**. Fluorescence lifetime images of GSBs and mammalian cells were collected using a Bergamo II multiphoton microscope equipped for digiFLIM (all digital time-correlated single-photon counting) coupled with a Tiberius tunable femtosecond Ti:Sapphire laser (both from Thorlabs Inc., Newton, NJ), with an Olympus UMPlanFL N 10x water immersion objective (NA 0.3). T-Sapphire-based biosensors were excited with 790 nm light, and emission light was split with an FF562-Di03 dichroic mirror and bandpass filtered for green (FF01-525/50) and red (FF01-641/75) fluorescence channels. Biosensors with mTurquoise2 were excited with 850 nm light, and the emission light was filtered with a FF01-482/35 bandpass filter (all filter optics from Semrock, Rochester, NY).

Hippocampal neurons were visualized with a 60× water-immersion objective (NA 1.0, Olympus LUMPLFLN) and LiLac was excited at 850 nm using a Chameleon Vision-S tunable Ti:Sapphire mode-locked laser (80 MHz, ~75 fs pulses, Coherent, Santa Clara, CA). Fluorescence emission was split with an FF562-Di03 dichroic mirror, bandpass filtered (FF01-482/35 filter), and detected with a hybrid photodetector R11322U-40 (Hamamatsu Photonics, Shizuoka, Japan). The photodetector signals and laser sync signals were preamplified and digitized at 1.25 GHz using a field-programmable gate array board (PC720 with FMC125 and FMC122 modules, 4DSP, Austin, TX).

Lifetime histograms were fitted using nonlinear least-squares fitting in MATLAB (Mathworks, Natick, MA), with a two-exponential decay convolved with a Gaussian for the impulse response function[47,58]. "Empirical lifetime" values, used for pseudocoloring the image data, are calculated as the mean photon arrival time minus the fitted value for $t_0$. For experiments in cultured cells and ex vivo preparations, lifetime values are reported as a standardized "$\tau_8$" value, where restriction of the averaging to the approximate time window of the actual data (0–8 ns) minimizes differences between experimental setups[10].

**Purified biosensor analysis in GSBs**. GSBs can be prepared with purified biosensor protein. For this type of preparation, purified Peredox biosensor protein was

first stripped of NADH by dialysis with 1 mM NAD$^+$, 10 mM pyruvate, 5 U/mL LDH (Worthington Biochemical Corp., Lakewood, NJ) in 50 mM MOPS, pH 7.3, 90 mM KCl, 10 mM NaCl. It was then added to a mixture of 1% agarose gel (in sol; Type IX-A, Sigma) and 1% alginate (Pronova UP LVG, Dupont), and emulsified at 30 °C using a droplet generator. 5 μL of gel droplets were dispersed in 500 μL of a polydisperse PAH emulsion (10 mg/mL PAH, Alfa Aesar 43092, MW~120,000 in 500 mM NaCl, vortexed in HFE7500 + 0.15% RAN007 fluorinated surfactant) and the mixed emulsion broken with 30% PFO ($^1$H,$^1$H,$^2$H,$^2$H-perfluoro-1-octanol; VWR, B20156-18) to allow polymer complexation and shell formation. The resulting GSBs contain only the purified biosensor protein.

GSBs were deposited on glass coverslips and imaged on 2p-FLIM while perfusing conditions prepared in 50 mM Tris, 90 mM KCl, 10 mM NaCl, with the indicated pH and ligand concentration. Peredox conditions were prepared as previously described[47]. Briefly, different ratios of lactate/pyruvate were used to produce different NAD$^+$/NADH ratios by adding purified LDH, a highly reproducible method used as an adjustment for reagent impurities. Experimental solutions contained 1 mM total NAD (NADH and NAD$^+$ forms) and 10 mM total lactate and pyruvate. Dose-response data were analyzed using GraphPad Prism 7.02[47].

**Library generation**. TlpC-TS and -TQ libraries sampled 49,152 unique biosensor variants with different linker compositions, lengths, and insertion points. Libraries were generated using degenerate oligonucleotides containing codon variation within the linker regions, flanked by an annealing sequence and a Gibson overhang sequence (Genewiz). Degenerate oligos (TlpCTS_LL01_F: CAT-AAGCTTGAGTACAACTYCWMCGSCSR-CRYCGGCATTGACCCCTTTACTGAA, TlpCTS_LL01_R: GTTTGTCAGCCATGATATAAACGTTGYDGKSGYK-GRAAAAGACTAAAGATTTATTG; TlpCTQ_LL01_F: CTGGGGCA-CAAGCTGGAGTACAACTYCWMCGSCSR-CRYCGGCATTGACCCCTTTACTGAA, TlpCTQ_LL01_R: CTTGTCGGCGGTGATATAGACGTTGYDGKSGYK-GRAAAAGACTAAAGATTTATTG) were used as primers in a standard PCR reaction (Q5 Hot Start High-Fidelity 2X Master Mix, NEB) to amplify T-Sapphire- or mTurquoise2-containing template (pRsetB-Peredox or Lck-cpmTq2-Calcium-lifetime-sensor, respectively).

To prepare the vector backbone, we first generated a staging plasmid consisting of a pRsetB backbone, E. coli codon-optimized T-Sapphire or mTurquoise2 fluorescent protein, and an E. coli codon-optimized TlpC gene (synthesized as a gBlock, IDT) inserted at position 144 within the fluorescent protein. The staging plasmid was either digested with HindIII and AclI for the TlpC-TS library, or PCR amplified for the TlpC-TQ library (BB_TQ_F: GTCTATATCACCGCCGAC, BB_TQ_R: GTTGTACTCCAGCTTGTG).

Plasmid libraries were generated by 2-fragment Gibson assembly of the library insert and the vector backbone (NEBuilder HiFi DNA Assembly, New England Biolabs). To eliminate incomplete reaction products, the Gibson product was "polished" by PCR using vector-specific primers (pRSET_F: cgcgttggccgattcatt, pRSET_R: gaagcatttatcagggttattgtctcatg), yielding a linear DNA library containing critical elements for transcription and translation (T7 promoter, RBS, and T7 terminator).

**Emulsion PCR**. Single copy template DNA was isolated and amplified in microfluidic droplets by performing emulsion PCR. Dilute "polished" template DNA (0.1 pg/μL of 2 kb template) was mixed with PCR reagents and emulsified at a rate of 10$^7$ droplets/hour using a microfluidic droplet generator designed for producing 35 μm droplets (Droplet Genomics, DG-DM-35). The reaction also included hot-start PCR reagents (Q5 Hot Start High-Fidelity 2X Master Mix, NEB), 0.05 μM 5'-DualBiotin-Reverse primer (5'-DualBiotin-iSp18-TGAAG-CATTTATCAGGGTTATTGTCTCATG, IDT; iSp18 is an internal 18-atom hexa-ethyleneglycol spacer), and 0.5 μM 5'-TexasRed-Forward primer (5'-TexasRed-CGCGTTGGCCGATTCATT, Genewiz). The DualBiotin-Reverse primer allows downstream immobilization on streptavidin beads, and anneals ~300 bp downstream of the T7 Terminator in order to distance the open reading frame from the microbead surface. The TexasRed fluorophore allows confirmation of DNA loading onto microbeads. The 600 μL droplet emulsion was divided among PCR tubes (50 μL/tube), overlaid with mineral oil, and thermocycled (PCR program: 98 °C, 30 s; [98 °C, 10 s; 60 °C, 16 s; 72 °C, 1 min 30 s] x 30; 72 °C, 2 min; 4 °C, hold) without a heated lid. Following PCR amplification, the emulsion was pooled, gently spun in a picofuge, and the fine emulsion recovered from beneath the coarse emulsion.

To confirm amplification, a diagnostic sample of the emulsion was stained with SYBR Green (Sigma-Aldrich) and imaged. Based on the Poisson distribution and an average of 1 copy of template per droplet, one would expect 37% of the droplets to have no amplification (0 copies of template) and 63% to have amplification as shown by SYBR Green fluorescence (≥1 copy of template).

**Capture of clonally-amplified DNA on affinity beads**. To maintain clonality, streptavidin beads were introduced microfluidically into individual PCR droplets

via controlled active droplet merging[59], then recovered from the droplets, washed, and concentrated.

To minimize protein aggregation during the downstream IVTT reaction, each streptavidin bead was limited to ~100,000 copies of DNA by pre-blocking a subset of streptavidin binding sites with biotin. 750 µL of streptavidin beads (6–8 µm diameter NeutrAvidin coated polystyrene particles at $3.4 \times 10^4$ beads/µL; Spherotech, NVP-60-5) were incubated with 3 µL of 1 µM of 5'-DualBiotin-Reverse primer for 5 min, then washed three times with Binding Buffer (1 M NaCl, 0.5 mM EDTA) using Pierce™ Spin Cups. Finally, the beads were resuspended in 400 µL of 1.5X Binding Buffer (1.5 M NaCl, 0.75 mM EDTA). The high salt concentration was density matched to the polystyrene particles to prevent bead settling.

PCR droplets were reinjected into a custom droplet merging device (Droplet Genomics) and synchronized by size-dependent flow with streptavidin bead droplets generated on-chip. Paired droplets were electrocoalesced at a rate of $10^6$ droplets/hour (Suppl. Fig. 2) when they pass an electrode delivering a high voltage, high-frequency pulse (10 kHz, 0.5 kV output). The voltage was applied using a Digilent Discovery 2 Pulse Generator connected to a Trek Model 2220 High Voltage Amplifier, and the electrical connection made with high voltage cables (Pasternack, PE3C3334-200CM).

The merged emulsion was collected off-chip in an Eppendorf tube and subsequently broken with 35% PFO ($^1H,^1H,^2H,^2H$-perfluoro-1-octanol; VWR, B20156-18) in the presence of 1 M NaCl, 0.5 mM EDTA, 1 mM biotin. Beads were washed three times with 1 M NaCl, 0.5 mM EDTA using Pierce™ Spin Cups (VWR 69702), and resuspended to a final density of ~200,000 beads/µL in 5 mM Tris, pH 7.3, 9 mM KCl, 1 mM NaCl for immediate use with IVTT, or in TE for long term storage at 4 °C (with a mineral oil overlay to prevent evaporation). To confirm DNA loading onto the beads, a diagnostic sample was evaluated for TexasRed fluorescence using epifluorescence imaging.

**Library expression in microfluidic droplets**. To express biosensor proteins in droplets, DNA beads were encapsulated in droplets containing purified IVTT reagents (PUREfrex2.0, GeneFrontier) supplemented with chaperones (DnaK mix, GeneFrontier).

To prevent premature transcription, DNA beads were microfluidically delivered into the IVTT reagents immediately prior to droplet encapsulation using a custom two-stream co-flow droplet generator device (Droplet Genomics). Flow rates were controlled so that the bead suspension was combined in a 1:4 ratio with the IVTT reagents to minimize reaction dilution. For a total reaction volume of 40 µL, 8 µL of DNA beads and 32 µL of IVTT reagents (20 µL Solution I, 2 µL Solution II, 2 µL Solution III, 2 µL DnaK mix, 6 µL RNase-free water) were injected into the co-flow droplet generator at a rate of 25 µL/hr and 100 µL/hr, respectively. Droplet generation oil was injected at 800 µL/hr to achieve a 30 µm droplet size. IVTT droplets were collected off-chip and biosensor protein expressed overnight at 30 °C.

**GSB preparation**. IVTT droplets were transformed into GSBs via a series of droplet manipulation steps. First, agarose (Type IX-A agarose, Sigma) and alginate (Pronova UP LVG and Pronova UP VLVG alginates, Dupont) were delivered into the IVTT droplets by electrocoalescence. To prevent premature gelation of the agarose, gel handling and droplet merging were performed in a temperature-controlled room held at 30 or 37 °C.

Stocks of 5% agarose (in sol), 5% VLVG alginate, and 3% LVG alginate were prepared in 50 mM HEPES, pH 7.5, 90 mM KCl, 10 mM NaCl and combined with Ni-NTA nanospheres (Kisker-biotech, PPS-0.2NI-NTA) and sterile 50% glycerol to final concentrations of 1.5% agarose, 1.5% LVG alginate, 0.4% VLVG alginate, 5% glycerol, and 2.5 mg/mL Ni-NTA nanospheres in 200 µL. The Ni-NTA nanospheres help retain His-tagged biosensor proteins during downstream polymer deposition, while the glycerol was added to match the osmolarity of the IVTT droplets. The gel/polymer mixture and IVTT droplets were injected onto a droplet merging device at 25 µL/hr and 20 µL/hr, respectively, with 165 µL/hr droplet generation oil and 50 µL/hr spacing oil. Paired IVTT and gel/polymer droplets were merged using the same pulse parameters as for PCR and bead droplets. The merged emulsion was collected off-chip, then put on ice for 10–20 min to gel the agarose.

To coat the gel droplets with the polyelectrolyte shell, 5 µL of droplets were suspended in 500 µL of polydisperse PAH emulsion, and both emulsions broken with 30% PFO. The resulting GSBs were pelleted with gentle centrifugation and washed three times with 50 mM Tris, pH 7.5, 90 mM KCl, 10 mM NaCl. Concentrated GSBs were deposited on nitric acid-etched, ethanol-flamed coverslips (Square Cover Glass, #1.5, 10×10mm, Ted Pella), treated with 50 mM EDTA for 20 min to release His-tagged biosensor protein from the Ni-NTA nanospheres, and immediately screened for biosensor responses.

**Sensor screening in GSBs**. Coverslips coated in GSBs expressing a biosensor library were placed in a perfusion chamber and imaged at ambient temperature with 2p-FLIM. Conditions were prepared in 50 mM Hepes, pH 7.5, 90 mM KCl, 10 mM NaCl, and different conditions were exchanged between imaging sets, sampling a total of 6–10 conditions during an experiment. Fluorescence lifetime images were collected using the ThorImageLS software (v4.2.2020.3061) and analyzed using laboratory-built software written in MATLAB. Tiled fluorescence

lifetime images were collected for the full coverslip; the first series was analyzed to find all circular ROIs within a selected brightness range (using the stock function imfindcircles.m); and the full image series was analyzed to give dose-response curves for each individual ROI. For all dose responses, solutions with different lactate concentrations were delivered in a randomized order to minimize photo-bleaching artifacts.

**Genotype recovery from single GSBs**. After screening and identifying winners, a thin-walled capillary was used to retrieve target GSBs. Retrieval capillaries were pulled from borosilicate glass capillary tubes on a horizontal heated-filament puller (P-97; Sutter Instrument) to a tip diameter of 15–30 µm, manipulated using a Burleigh PCS-6000 (Thorlabs), and attached to a syringe via plastic tubing to allow manual control of suction. Before immersing the capillary into the bath solution, the capillary was backfilled with 2–5 uL of 0.1 M NaOH. The tip was positioned next to the target GSB, and the 0.1 M NaOH solution slowly dispensed to gently dissolve the polyelectrolyte shell. Then, the exposed gel and DNA bead were pulled into the capillary tip, removed from the bath, and delivered into a PCR tube containing 4 µL of 10 mM Tris, pH 7.3.

PCR reagents were added to the tube (Q5 Hot Start Master Mix, NEB) with 0.5 µM insert-specific forward and reverse primers, and thermocycled (98 °C, 30 s; [98 °C, 10 s; 50 °C or 64 °C, 16 s; 72 °C, 30 s] x30; 72 °C, 2 min; 4 °C, hold). For TlpC-TS GSBs, the recovery primers (TlpC_TS_Recovery_F: TTACTACCTTCT CCTATG, TlpC_TS_ Recovery_R: TACCGTTCTTCTGTTT) were annealed at 50 °C to yield a 1104 bp amplicon. For TlpC-TQ GSBs, the recovery primers (TlpC_TQ_ Recovery_F: CATCGAGCTGAAGGGCATC, TlpC_TQ_ Recovery_R: GTCCTCGATGTTGTGGC) were annealed at 64 °C to yield a 964 bp amplicon. A portion of the gel-purified DNA was Sanger sequenced immediately to recover the identity of the linker regions, while the remainder of the amplicon was reintegrated into a vector for downstream validation and cloning.

Backbone DNA was prepared by PCR amplifying the corresponding staging plasmid (TlpC-TS primers: GATAGCGGGCGAAAAC, AACGTTTATATCATG GCTGAC; TlpC-TQ primers: GCCACAACATCGAGGAC, GATGCCCTTCAGC TCGATG). Backbone DNA was gel-purified and combined with the insert DNA using Gibson assembly (NEBuilder HiFi DNA Assembly, New England Biolabs). NEB5alpha cells (NEB) were transformed with the Gibson product, and individual colonies were cultured, miniprepped and sequenced. Biosensors were also subcloned into bacterial expression vectors so the biosensor protein could be purified and subjected to in vitro specificity testing.

**Uniform biosensor analysis in IVTT-GSBs**. To prepare a uniform sample of LiLac GSBs, pre-blocked streptavidin beads were bulk loaded with purified LiLac DNA that was PCR amplified from a single plasmid with 5'-TexasRed-Forward and 5'-DualBiotin-Reverse primers. Uniform DNA beads were used to generate a uniform sample of GSBs, following the same protocol as for biosensor libraries.

Dose-response data were analyzed using GraphPad Prism 7.02. Data were plotted as fluorescence lifetime (ns) against [lactate] (mM) and fit with a Hill equation with the form of Eq. 1:

$$LT = LT_{min} + \frac{LT_{max} - LT_{min}}{1 + \left( \frac{[lactate]}{K_{0.5}} \right)^{n_H}} \qquad (1)$$

where $LT_{min}$ and $LT_{max}$ are the lower and upper lifetime asymptotes, respectively, $K_{0.5}$ is the midpoint of the curve and $n_H$ is the Hill coefficient, which was fixed to −1.

For experiments performed at 34 °C, the bath temperature was controlled using an inline solution heater (64-0103, Warner Instruments, Hamden, CT) connected to a TC-324B temperature controller (Warner Instruments). The thermistor readout was monitored using ThorSync software (v4.0.2019.6171) and verified and corrected using a thermocouple biosensor linked to a BAT-12 thermometer (Sensortek, Clifton, NJ).

**Protein expression and purification**. *Escherichia coli* BL21 DE3 (NEB) were transformed with pRsetB-LiLac. A single colony was used to inoculate 2 1-L cultures in Terrific Broth, overnight at 37 °C, 200 rpm. The cultures were cooled on ice, supplemented with 0.1 mM IPTG, and induced at 18 °C, 200 rpm for 36 h. Pellets were centrifuged, snap-frozen in liquid nitrogen, and stored at −80 °C.

Frozen cell pellets were re-suspended in ice-cold lysis buffer (50 mM $NaH_2PO_4$, 10 mM imidazole, pH 8), supplemented with 1 tab cOmplete EDTA-free protease inhibitor cocktail (Roche) per 25 mL buffer, 1 mM PMSF, 0.1 mg/mL lysozyme, and 1 µL Pierce Universal Nuclease (Thermo Fisher) per 25 mL buffer. The re-suspension was sonicated on ice at 30% power, in pulses of 10 s on/20 s off, for a total of 2 min of sonication; the lysate was nutated for 30 min at 4 °C. The lysate was centrifuged for 20 min at 4 °C, 48,380 × g. The supernatant was flowed over Ni-NTA pre-equilibrated in lysis buffer, washed with 10 column volumes (CVs) of lysis buffer, and then washed with 10 CVs of wash buffer (50 mM $NaH_2PO_4$, 25 mM imidazole, pH 8). The protein was eluted in a buffer containing 50 mM $NaH_2PO_4$, 500 mM imidazole, pH 8. Concentrated eluted protein was further purified via size-exclusion chromatography, using a TSK G3000SWxL column (GE Healthcare) equilibrated in a running buffer consisting of 50 mM NaHepes, 150 mM NaCl, 2 mM $MgCl_2$, 0.1% CHAPS, 1 mM DTT, pH 7.4. Protein-containing fractions were pooled, concentrated using a 30 kDa MWCO spin

concentrator (Amicon), exchanged into a buffer containing 25 mM MOPS, 90 mM KCl, 10 mM NaCl, supplemented with 5% glycerol, flash-frozen in liquid nitrogen, and stored at −80 °C.

**Photophysical characterization**. Fluorescence excitation and emission spectra were acquired using a SpectraMax iD5 Multi-Mode Microplate Reader (Molecular Devices). LiLac was diluted to ~500 nM in MOPS buffer at room temperature with 0 or 100 mM lactate, and spectra were acquired in quadruplicate and then averaged. Excitation spectra were measured in 2 nm increments from 350–460 nm, measuring emission at 500 nm. Emission spectra were measured in 2 nm increments from 465–649 nm using excitation at 425 nm.

For determining the LiLac relative quantum yield, fluorescence measurements were made using a PTI QuantaMaster-8000 spectrofluorometer controlled by PTI Felix 32 software (Photon Technology International). Protein was diluted to $0.0025 < A_{440} < 0.025$ in MOPS buffer at room temperature with or without lactate, in triplicates; acridine orange in ethanol with 0.01 M KOH was used as a standard[60]. Fluorescence emission spectra were recorded from 450–650 nm (2.5 nm slit), using 440 nm excitation. Spectra were integrated and analyzed in GraphPad Prism.

For determining the LiLac extinction coefficient, absorbances were measured using a BioMate 160 UV-Vis spectrophotometer (ThermoFisher Scientific), using a range from 290 to 650 nm, with 2 nm step size and using buffer as a reference. Protein concentrations were measured using the Bio-Rad Protein Assay (Bio-Rad, Hercules). Extinction coefficients in lactate-free and lactate-bound states were determined using the Beer-Lambert Law, in technical triplicates.

**In-cell calibration of LiLac**. The in-cell calibration of LiLac was performed as previously described[10]. Briefly, permeabilized HEK293T cells (obtained from ATCC) expressing the LiLac biosensor were imaged in the presence of different concentrations of lactate delivered in the following base solution (in mM): 140 K-Gluconate, 10 NaCl, 10 HEPES, 1 EGTA, 1.324 MgCl₂, 0.346 CaCl₂, and pH 7.35 at 34 °C. Cells were transferred to the imaging chamber, and after ~10 min in the solution, were permeabilized with 45 μM β-escin for ~2 min, and then imaged in the solution without the permeabilizing agent. All solutions contained 2 μM of the lactate dehydrogenase inhibitor, GSK-2837808A (Tocris Bioscience, Cat#5189; CAS:1445879-21-9).

**Sensor performance in mammalian cells**. HEK293T cells were cultured in a 37 °C 5% CO₂ environment in DMEM supplemented with Glutamax, 10% fetal bovine serum, and penicillin/streptomycin. For transfection experiments, ~150,000 cells/well were seeded in 6 well plates containing clean, protamine-coated glass coverslips. After reaching ~50% confluence, the cells were transfected with ~500 ng/well pAAV-CAG-LiLac, using the polyethylenimine method. Cells were imaged 16–48 h post-transfection. Cells were transferred to pre-warmed 0 glucose-Hepes-ACSF (10 mM Hepes, 1 mM NaH₂PO₄, 146 mM NaCl, 2.5 mM KCl, 2 mM CaCl₂, 1 mM MgCl₂, pH 7.4) 20 min prior to 2p-FLIM imaging at 34 °C.

The signal-to-noise ratio (SNR) is calculated as signal/noise. Signal was estimated by taking the mean lifetime values in cells at 0 and 10 mM external lactate, and then taking the absolute difference between the two conditions. Noise was estimated by computing the standard deviation of values for each condition and then averaging. Data shown in Suppl. Fig. 6.

**Mice**. Male and female wild-type C57BL/6 N mice of between 14 and 24 days old were used in this study. Animals were bred in-house in ventilated cages within a barrier facility, which maintained 12 hr light/dark cycle, regulated cage temperature (24 °C) and humidity (53%) and provided ad libitum access to water and food (Picolab Rodent Diet 5053). All experiments were performed in compliance with the NIH Guide for the Care and Use of Laboratory Animals and Animal Welfare Act. Specific protocols were approved by the Harvard Medical Area Standing Committee on Animals.

**Sensor expression in the brain**. Custom-made adeno-associated vectors (AAV; obtained from the Viral Core Facility in Boston Children's Hospital or the NYUAD viral core) were used for biosensor expression in acute brain slices. For expression of Laconic we used AAV2/9 serotype and CAG promoter. For expression of LiLac we used PhP.eB serotype and CAG promoter.

Laconic or LiLac expression in the hippocampus was achieved by intracranial stereotactic injection of P1-P2 mouse pups using published methods[61]. Following cryoanesthesia, pups were injected with 150 nl of AAV, twice per hemisphere, at the following coordinates with respect to lambda: (i) 0 mm in the anterior-posterior direction, ±1.9 mm in the medial-lateral axis, and −2.0 mm in the dorsal-ventral direction; and (ii) 0 mm in the anterior-posterior direction, ±2.0 mm in the medial-lateral axis and −2.3 mm in the dorsal-ventral direction.

**Hippocampal Slice Preparation and Imaging**. Mice between 14 and 24 days old were anesthetized with isoflurane, decapitated, and the brain was removed into ice-cold slicing solution containing (in mM) 87 NaCl, 2.5 KCl, 1.25 NaH₂PO₄, 25 NaHCO₃, 75 sucrose, 25 D-glucose, 0.5 CaCl₂, and 7 MgCl₂ (~335 mOsm/kg) and

bubbled with 95% O₂ and 5% CO₂. The brain was glued by the dorsal side, embedded in 2% agarose in phosphate-buffered saline, and submerged in a chamber with ice-cold slicing solution. Horizontal 275 μm slices were cut using a Compresstome (VF-310-0Z, Precisionary, Natick, MA) and immediately transferred to a chamber with artificial cerebrospinal fluid (ACSF) containing (in mM) 120 NaCl, 2.5 KCl, 1 NaH₂PO₄, 26 NaHCO₃, 10 D-glucose, 2 CaCl₂, and 1 MgCl₂ (~290 mOsm/kg) that was warmed to 36 °C and bubbled with 95% O₂ and 5% CO₂. After 35 min, the chamber was moved to room temperature and slices therein were used for the next 4 h. Slices were adhered to poly-L-lysine-coated coverslips, placed in a bath chamber mounted on a 2p-FLIM microscope, and perfused with ACSF bubbled with 95% O₂ and 5% CO₂. Sodium-lactate was dissolved directly in ACSF solutions. Glucose-free ACSF was made by omitting D-glucose from the solution. The temperature was controlled using an inline solution heater (64-0103, Warner Instruments, Hamden, CT) connected to a TC-344C temperature controller (Warner Instruments). Microscope control and image acquisition were performed by a modified version of the ScanImage software written in MATLAB[62].

**Reporting summary**. Further information on research design is available in the Nature Research Reporting Summary linked to this article.

## Data availability

Numerical source data for all of the figures are provided in the Source Data file; the DNA and protein sequences for the LiLac sensor are provided in Supplementary Note 1.

Plasmids encoding LiLac have been deposited and are distributed through addgene.org (pRsetB-KanR-LiLac as Addgene #184569 and pAAV-CAG-LiLac as Addgene #184570); plasmid maps and sequences can be downloaded from the addgene.org website.

## Code availability

Code for analyzing bead images is publicly available at https://github.com/gyellen/BeadScan.

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

## Acknowledgements

We thank members of the Yellen lab and M. Tantama and J. Markowitz for helpful discussions and comments on the manuscript, M. Rahman, N. Nathwani and J.R. Martínez-François for expert technical assistance, M. Valenstein for assistance with FPLC, and T. Kula and B. Powell for fluorometer access. We thank the Harvard Medical School Neurobiology Research Instrumentation Core Facility and the Neurobiology Machine Shop (both supported by NIH grant P30 EY012196) for help building the 3D printed syringe pumps. We also thank J. Nainys for assisting with custom microfluidic device design (Droplet Genomics); and the Viral Core of Boston Children's Hospital, and System Biology, NYU, Abu Dhabi, UAE (as well as Dr. Gordon Fishell) for packaging of AAVs. The Laconic biosensor was a gift from L. Felipe Barros (Addgene plasmid #44238), and Lck-cpmTq2-Calcium-lifetime-sensor was a gift from Dorus Gadella (Addgene plasmid #129627). This work was supported by NIH grant R01 GM124038 (to G.Y.) and NIH fellowships F31 CA254162 (to P.C.R.), F32 NS116105 (to D.J.M.), and F32 GM123577 (to D.K.).

## Author contributions

G.Y., D.K., P.J.C., D.A.W., and L-H.C. conceptualized the study. D.K. performed the experiments, analyzed the data, and wrote the manuscript. P.C.R., C.M.D-G., and D.J.M. performed experiments and contributed to the writing of the manuscript. D.K., L-H.C., and Y.W. performed experiments to establish project feasibility. G.Y., D.K., P.C.R and D.J.M. acquired funding. G.Y. supervised the study, wrote the software, and wrote the manuscript. All authors discussed the results and commented on the manuscript.

## Competing interests

The authors declare no competing interests.

**Additional information**

