## [Peer Review File · Nature Communications]

REVIEWER COMMENTS

Reviewer #1 (Remarks to the Author):

In this truly original and exciting work by Koveal D. and colleagues, the authors describe a novel bead-based high-throughput screening assay for the multiparametric screening of genetically encoded biosensor libraries. To achieve this feat, the authors implemented a clever new method involving PCR-based amplification of DNA in individual beads followed by affinity capture of the DNA product in individual beads for in vitro transcription/translation reactions that lead to high-yield protein production in an encapsulated microenvironment. These beads can then be subjected to screening for a variety of parameters under perfusion conditions on a glass coverslip.

The authors field-test their new screening method with the development of a new sensor for Lactate. First they tried using T-Sapphire, but the resulting sensor variant was still responsive to pH. Then they repeated their screening using mTurquoise2 as fluorescent protein. They isolated a variant that is insensitive to pH but senses Lactate quite well in the physiological range. I believe this new sensor will be a game-changer in the field.

Overall the research work is of extremely high quality and originality, the results appear very solid and the newly developed Lactate biosensor will be undoubtedly very useful to several other laboratories. Furthermore the experimental planning was carefully-conducted, the project well-thought out and the manuscript well written. The findings are of very high importance and immediate relevance to the field. I only have a few minor comments, listed below. Upon consideration of these, I fully endorse this publication.

Minor comments:

1. At page 6, the authors compare the SNR of LiLac and Laconic, with a specific subscript underlining that they are using Laconic in lifetime mode. The most common metric of SNR in FLIM is simply the square root of the total number of collected photons. In that case, their statement would be equivalent to saying that Lilac is about 625 times brighter than Laconic under identical illumination conditions, which seems unlikely based on photophysical properties only. Therefore, either this is due to vastly different expression levels, or the authors must have used a different metric, such as a relation between the lifetime value and its variation under some standard “baseline” conditions. It would be a valuable addition to specify in the methods section how this metric was calculated.

2. The authors chose the lactate-binding protein TlpC as a protein scaffold for sensor development. This protein belongs to a class of bacterial chemoreceptors that are known to possess two ligand-binding domains (PAS-like domains), one being membrane-proximal and one membrane-distal in the original bacterial protein (see Mayra A. Machuca et al, Scientific Reports 2017). While the lactate-binding domain is well characterized the second ligand binding domain might have an unknown ligand that could as well trigger conformational change that may or may not trigger a response in the biosensor. While I appreciate the effort the authors did by testing LiLac responses to a small subset of molecules for evaluating the sensor's specificity, we cannot exclude that in mammalian cells LiLac might also respond to a second ligand of unknown nature (perhaps an aminoacid). I think that it would be important that the authors would at least discuss this aspect in the paper.

3. A more appropriate title for this work should be: “A high-throughput multiparameter screen for accelerated development and optimization of soluble genetically encoded fluorescent biosensors”, since as is mentioned in the discussion this new approach is only suitable for biosensors that can be expressed in soluble form.

4. The authors refer to membrane-associated and GPCR-based sensors without references. References should be added.

5. It would be nice if the authors could mention in the discussion if they envision that their new assay will be easy or difficult to implement in other labs

6. In the methods section under “Sensor performance in mammalian cells” there’s a typo in the first line: “37 5% CO₂” (the Celsius degree symbol is missing).

Reviewer #2 (Remarks to the Author):

Koveal and coworkers describe a technologically sophisticated and effective system (BeadScan) for screening libraries of genetically encoded biosensors. The traditional methods for biosensor screening are to express libraries of biosensors in bacterial or mammalian cells and screen either the extracted protein or the whole-cell response. This is labor intensive and, generally speaking, there is a limit to the number of parameters that can be screened simultaneously. Koveal and coworkers have attempted to overcome the limitations of traditional approaches by using a cell-free system for biosensor expression and screening. The key to their approach is the use of gel-shell beads (GSBs) that are formed using a microfluidic system and can be immobilized on glass coverslips where they are imaged. Particular clones of interest can be picked off of the coverslip using a microcapillary and the DNA can be recovered for downstream applications by PCR.

I find this work to be aptly described as a technological tour-de-force. While I’m sure that many protein engineers have dreamt of performing cell-free screening, most would have been scared off by the sheer number of technical challenges that would need to be overcome. Indeed, Koveal and coworkers appear to have solved a multitude of technical challenges to get their system to work as well it would. This brings me to my first major concern with this manuscript, which is how tersely the actual BeadScan system is described in the main text. The description in the main text does not do justice to the technological sophistication of the system, and leaves out too many important details. I feel that the main text should be expanded to more thoroughly describe the BeadScan system, with particular emphasis on describing how the main technological challenges were overcome. Two examples of small details that might deserve mention are the biotin blocking step before DNA immobilization, and the use of the Ni-NTA nano spheres during GSB preparation.

The use of the BeadScan system led to the development of a new lactate biosensor (LiLac) optimized for FLIM imaging. My second major concern with this manuscript is that LiLac has not been thoroughly

characterized in terms of its photophysical parameters. At a minimum, it is standard practice to measure molecular brightness (quantum yield and extinction coefficient) and to provide absorbance and fluorescence spectra for a soluble biosensor. More detailed characterization could include 2P cross-section, photostability and brightness in mammalian cells.

My third and final major concern is that I feel the Discussion section could be a bit more detailed and thorough in terms of the pros and cons of BeadScan relative to alternative approaches. For example, the actual libraries screened in this work (~1000s) are similar to what could be achieved using manual picking of colonies into 96-well plates, within a couple of weeks. What is the time required for one round of BeadScan? How about the cost to set up and run? Doesn't the use of IVTT mean that the proteins are expressed in a very unnatural environment, and so folding and other properties may not translate to the cellular milieu? How might the system itself be further improved or streamlined?

Despite my concerns, I feel that this is an impressive and important contribution. I recommend that this work could be acceptable for publication once the major concerns listed above, and additional concerns listed below, have been addressed.

Additional concerns to be addressed.

- The procedure requires four distinct microfluidic steps, the first two of which could presumably be combined into one step if the biotinylated primers were pre-immobilized to the beads. The authors state that pre-immobilization would limit the number of DNA copies per bead, but it is not clear from the data provided in this manuscript that this is truly a limiting factor or an important consideration. While the authors mention that pre-immobilized primers could limit amplification efficiency due to crowding, couldn't the same argument be applied to the efficiency of transcription of more densely modified beads? There must be some trade-off here, and the authors have not convinced this reader that they have explored these parameters thoroughly enough to draw strong conclusions about the optimal approach.

- On page 30, it is stated that, "Biosensors were also subcloned into bacterial expression vectors so the biosensor protein could be purified and subjected to in vitro specificity testing." However, further experimental methods of bacterial expression, purification, and methods for characterization of purified proteins, seem to be missing. Furthermore, it is not exactly clear which in vitro characterization experiments were done with purified proteins, and which were done with IVTT-produced LiLac in GSBs.

- The methods explains that beads were limited to ~100,000 copies of DNA per bead by pre-blocking with biotin. However, in the main text, it is stated that beads have >200,000 copies.

- I feel that it would be more appropriate to put actual measured lifetimes on the pseudocoloured bar in Figure 5, rather than "High" and "Low".

- The abstract mentions, "specificity" as a feature that can be evaluated using BeadScan, but that was not demonstrated in this work.

- In the Discussion, statements about the range of conditions that can be evaluated (e.g., "can be assayed against many conditions") would be toned down a bit and it should be made clear that BeadScan could "potentially" enable this.

Reviewer #3 (Remarks to the Author):

Koveal and co-authors developed BeadScan, which is a screening method for developing fluorescent biosensors using droplet microfluidics. They first created a DNA library that codes the protein biosensor. The droplet-isolated template DNA was amplified by PCR inside the droplet (with each droplet containing 0 or 1 DNA molecule). Then the droplet was fused with another droplet containing a bead. After breaking the droplet and washing out excessive DNA, the single DNA-containing bead was re-encapsulated in a droplet that contains purified IVTT (in vitro coupled transcription/translation). Eventually, after the expression of biosensor protein, the droplet transformed into gel-shell beads (GSB) which allows the passing of small molecules while trapping the protein biosensor inside. The GSBs adhered to the glass surface and substrate for the biosensor can be added to examine the biosensor performance in terms of fluorescence lifetime. Using the BeadScan method, the authors developed a biosensor (LiLac) that could quantitatively measure lactate concentrations in living cells and tissue. The authors demonstrated the advantage of LiLac by comparing it with the existing methods Laconic in terms of response size, variability, and signal-to-noise ratio. The work in the paper was well executed. However, the level of innovation is fairly low. The droplet microfluidic operation (droplet formation and fusion) is very routine. GSB approach is interesting but was developed in 2014 by a different lab for use in biocatalyst screening (see ref 17 in the manuscript). The application to biosensor screening is a fairly trivial change from the original 2014 work.

Specific comments:

1. On page 3, why use the microbead to capture amplified DNA when the authors can infuse the droplet directly with IVTT system? Was there any interference between the reagents and IVTT system?
2. On page 5, the authors need to clarify whether the lactate concentration and pH were changed in the same experiment for TlpC-TQ. If the tests of the two parameters were done sequentially, the authors need to state that.
3. On page 5, since mTurquoise2 was used to develop the fluorescent lifetime sensor for calcium level, and the authors mentioned on page 2 that one of the limitations for existing lactate biosensor was an undesirable response to calcium. The authors should test LiLac in response to the change of calcium level.
4. An important selling point of the manuscript seems to be on multiparameter screening. However, the screening against individual parameters seems to be done separately in their experiments. These tests were done sequentially (instead of simultaneously). Thus the claim on multiparameter screening is misleading.

REVIEWER COMMENTS *and author responses (italicized)*

Reviewer #1 (Remarks to the Author):

In this truly original and exciting work by Koveal D. and colleagues, the authors describe a novel bead-based high-throughput screening assay for the multiparametric screening of genetically encoded biosensor libraries. To achieve this feat, the authors implemented a clever new method involving PCR-based amplification of DNA in individual beads followed by affinity capture of the DNA product in individual beads for in vitro transcription/translation reactions that lead to high-yield protein production in an encapsulated microenvironment. These beads can then be subjected to screening for a variety of parameters under perfusion conditions on a glass coverslip.

The authors field-test their new screening method with the development of a new sensor for Lactate. First they tried using T-Sapphire, but the resulting sensor variant was still responsive to pH. Then they repeated their screening using mTurquoise2 as fluorescent protein. They isolated a variant that is insensitive to pH but senses Lactate quite well in the physiological range. I believe this new sensor will be a game-changer in the field.

Overall the research work is of extremely high quality and originality, the results appear very solid and the newly developed Lactate biosensor will be undoubtedly very useful to several other laboratories. Furthermore the experimental planning was carefully-conducted, the project well-thought out and the manuscript well written. The findings are of very high importance and immediate relevance to the field. I only have a few minor comments, listed below. Upon consideration of these, I fully endorse this publication.

We appreciate the reviewer's favorable evaluation of our work.

Minor comments:

1. At page 6, the authors compare the SNR of LiLac and Laconic, with a specific subscript underlining that they are using Laconic in lifetime mode. The most common metric of SNR in FLIM is simply the square root of the total number of collected photons. In that case, their statement would be equivalent to saying that Lilac is about 625 times brighter than Laconic under identical illumination conditions, which seems unlikely based on photophysical properties only. Therefore, either this is due to vastly different expression levels, or the authors must have used a different metric, such as a relation between the lifetime value and its variation under some standard "baseline" conditions. It would be a valuable addition to specify in the methods section how this metric was calculated.

The reviewer is correct that we have used a signal-to noise ratio (SNR) metric other than the square root of the collected photons. Here, we have used a SNR metric that estimates the magnitude of the lifetime signal in cells from 0 to 10 mM external lactate relative to the average noise level at each condition. Signal was estimated by taking the mean lifetime values in cells at 0 and 10 mM external lactate, and then taking the absolute difference between the two conditions. Noise was estimated by computing the standard deviation of values for each condition and then averaging. We have updated the description in the main text on pg. 7 and in the methods on pg. 22 to clarify this metric. We have also added the SNR values to the legend of Supplementary Fig. 6A–B, which presents the data from which these values were derived.

2. The authors chose the lactate-binding protein TlpC as a protein scaffold for sensor development. This protein belongs to a class of bacterial chemoreceptors that are known to possess two ligand-binding domains (PAS-like domains), one being membrane-proximal and one membrane-distal in the original bacterial protein (see Mayra A. Machuca et al, Scientific Reports 2017). While the lactate-binding domain is well characterized the second ligand binding domain might have an unknown ligand that could as well trigger conformational change that may or may not trigger a response in the biosensor. While I appreciate the effort the authors did by testing LiLac responses to a small subset of molecules for evaluating the sensor's specificity, we cannot exclude that in mammalian cells LiLac might also respond to a second ligand of unknown nature (perhaps an aminoacid). I think that it would be important that the authors would at least discuss this aspect in the paper.

As the reviewer notes, the distal PAS domain of TlpC was identified as a putative binding domain in the original structure paper for TlpC (Machuca et al 2017), where the authors hypothesized that the large shallow cleft may accommodate a peptide, or loop or terminal peptide of an-as-yet unidentified PBP. We appreciate the reviewer's point and have included a statement in the Discussion on pg. 8 highlighting this observation and qualifying that such an interaction may have an unknown effect on sensor output.

3. A more appropriate title for this work should be: "A high-throughput multiparameter screen for accelerated development and optimization of soluble genetically encoded fluorescent biosensors", since as is mentioned in the discussion this new approach is only suitable for biosensors that can be expressed in soluble form.

We are happy to include the proposed title change and have updated the title accordingly.

4. The authors refer to membrane-associated and GPCR-based sensors without references. References should be added.

We have added references to detailed reviews of GPCR-based sensors from Wang et al 2018 and Andreoni et al 2019.

5. It would be nice if the authors could mention in the discussion if they envision that their new assay will be easy or difficult to implement in other labs

We do anticipate that other labs will readily be able to implement the BeadScan system, and we have added a paragraph to the Discussion on pg. 9 comparing start up and run costs for the required microfluidics equipment relative to that needed for the more commonly used 96-well plate screening methods. We also recognize that many labs may be interested in screening for fluorescence intensity responses, not lifetime responses, and we believe that BeadScan could be adapted for screening on an epifluorescence microscope equipped with a motorized stage, instead of a 2p-FLIM system like the one used here.

6. In the methods section under "Sensor performance in mammalian cells" there's a typo in the first line: "37 5% CO₂" (the Celsius degree symbol is missing).

We thank the reviewer for their close reading of our manuscript, and we have fixed the indicated typo.

Reviewer #2 (Remarks to the Author):

Koveal and coworkers describe a technologically sophisticated and effective system (BeadScan) for screening libraries of genetically encoded biosensors. The traditional methods for biosensor screening are to express libraries of biosensors in bacterial or mammalian cells and screen either the extracted protein or the whole-cell response. This is labor intensive and, generally speaking, there is a limit to the number of parameters that can be screened simultaneously. Koveal and coworkers have attempted to overcome the limitations of traditional approaches by using a cell-free system for biosensor expression and screening. The key to their approach is the use of gel-shell beads (GSBs) that are formed using a microfluidic system and can be immobilized on glass coverslips where they are imaged. Particular clones of interest can be picked off of the coverslip using a microcapillary and the DNA can be recovered for downstream applications by PCR.

I find this work to be aptly described as a technological tour-de-force. While I'm sure that many protein engineers have dreamt of performing cell-free screening, most would have been scared off by the sheer number of technical challenges that would need to be overcome. Indeed, Koveal and coworkers appear to have solved a multitude of technical challenges to get their system to work as well it would. This brings me to my first major concern with this manuscript, which is how tersely the actual BeadScan system is described in the main text. The description in the main text does not do justice to the technological sophistication of the system, and leaves out too many important details. I feel that the main text should be expanded to more thoroughly describe the BeadScan system, with particular emphasis on describing how the main technological challenges were overcome. Two examples of small details that might deserve mention are the biotin blocking step before DNA immobilization, and the use of the Ni-NTA nano spheres during GSB preparation.

We appreciate the reviewer's favorable evaluation of our work. The reviewer is correct that we encountered a number of technical challenges as we established and optimized the BeadScan methodology, and we have further detailed our approach to overcoming these challenges in the revised main text on pgs. 3–4.

Regarding the DNA bead preparation, we found that strong expression of soluble sensor protein was achieved using beads that had been intentionally limited to ~100,000 copies of DNA by pre-blocking a subset of streptavidin binding sites. We tested beads of multiple sizes, carrying different payloads (e.g. various copy number of dsDNA of different lengths), and found that more densely loaded beads sometimes led to the accumulation of visible protein aggregates within the droplet (Supplementary Figure 1C). We reasoned that sparsely loaded beads may have fixed this problem by 1) spacing out DNA strands on the bead surface improving physical separation of transcripts and nascent polypeptide chains, and 2) altering the rates at which transcripts are produced and proteins are translated. We speculate that the distance between DNA strands is somehow a main driver of protein aggregation, as our optimization tests showed that fully-loaded 2 μ m beads delivering only ~10–20,000 copies/bead had more aggregates (nonuniform fluorescent punctae) than did sparsely-loaded 6 μ m beads delivering ~100,000 copies/bead (Supplementary Figure 1C). This result is counterintuitive to the expectation that aggregation should worsen at higher protein expression levels driven by a larger DNA payload. Although we have not deeply investigated the precise molecular underpinnings of these observations, we presume that the nascent polypeptide chains are not uniformly distributed in the droplet but rather concentrated around the DNA bead from which they are produced.

Ni-NTA nanospheres were included in the gels to retain sensor protein during the shell formation process. In earlier formulations omitting the Ni-NTA nanospheres, we estimated ~50-60% protein loss during the shelling process. This indicated that, upon emulsion disruption, the rate of protein diffusion out of the gel exceeded that of shell formation around the gel. Therefore, we included the Ni-NTA nanospheres to prevent outward protein diffusion during shell deposition. Once the shell has formed around the mature GSB, sensor protein is released from the nanospheres with mild EDTA treatment. This modification improved sensor retention to ~90-100% from IVTT droplets to mature GSBs. While this moderate improvement in sensor retention is important for sensors with lower expression levels or for dim sensors, the Ni-NTA nanospheres are an optional additive that could be omitted for very bright targets with high expression levels.

The use of the BeadScan system led to the development of a new lactate biosensor (LiLac) optimized for FLIM imaging. My second major concern with this manuscript is that LiLac has not been thoroughly characterized in terms of its photophysical parameters. At a minimum, it is standard practice to measure molecular brightness (quantum yield and extinction coefficient) and to provide absorbance and fluorescence spectra for a soluble biosensor. More detailed characterization could include 2P cross-section, photostability and brightness in mammalian cells.

We have now measured the quantum yield, extinction coefficient, and excitation/emission spectra for both the apo and the lactate-saturated holo state of the LiLac sensor (reported in Supplementary Table 1 and Supplementary Fig. 5).

My third and final major concern is that I feel the Discussion section could be a bit more detailed and thorough in terms of the pros and cons of BeadScan relative to alternative approaches. For example, the actual libraries screened in this work (~1000s) are similar to what could be achieved using manual picking of colonies into 96-well plates, within a couple of weeks. What is the time required for one round of BeadScan? How about the cost to set up and run? Doesn't the use of IVTT mean that the proteins are expressed in a very unnatural environment, and so folding and other properties may not translate to the cellular milieu? How might the system itself be further improved or streamlined?

We appreciate the reviewer's concern and have included new text to address these points in the Discussion on pg. 9. Briefly, while we found it necessary to screen just one coverslip displaying 1,411 GSBs to isolate LiLac, it is quite feasible to screen up to tens of thousands of GSBs in a week. It takes two days to prepare a sample that can yield up to 5 coverslips for screening, each with ~1-2,000 GSBs. This can easily be exploited should one wish to more thoroughly sample a library or need to search more extensively for very rare variants. Compared to 96-well plates, throughput of tens of thousands per week is a significant improvement over thousands in two weeks. Having performed both screening methods in our laboratory, we can also say from experience that GSB preparation is less taxing than manual colony picking, as the microfluidics do not require constant active monitoring.

Of course, the strongest advance comes from the ability to couple this higher throughput with multiparameter screening—co-evaluating affinity, response size, intensity, and specificity/pH responses. The costs of setting up the microfluidics necessary for BeadScan are also quite reasonable. Four of our microfluidic setups (each with an inverted microscope, high speed camera, four 3D-printed syringe pumps, one high voltage amplifier) can be built for the cost of one plate reader. Fully-integrated microfluidics systems are also commercially available for a moderately higher cost than

our system. Each droplet generator device costs about as much as a miniprep. These devices are commercially available, and because of the low volume requirement the total reagent costs are negligible. For these reasons, we believe the BeadScan system has the potential to be implemented by other interested labs.

Regarding the in vitro environment of the GSBs, the purified IVTT reagents used in our system offer a simplified and highly optimized means of protein expression. While small protein targets are expressed most efficiently, we have successfully expressed large sensors (up to ~80 kDa) that are fully functional. If protein folding is a problem, purified chaperones can be added to the system. As noted in the Methods, we routinely include the DnaK chaperone as an additive. In a sense, DnaK is added prophylactically, as we have not specifically identified protein folding as an issue for any of our targets. As for translating to the cellular environment, LiLac serves as an excellent example of how well a BeadScan-optimized sensor can translate into in vivo systems. We believe that LiLac's success arises from how thoroughly we were able to optimize its lactate response, making it robust against other potentially confounding factors in the cellular milieu. We anticipate that other soluble sensors subjected to equally stringent screening would also translate well from BeadScan to cells. We have added new text touching on these points in the Discussion on pg. 9.

We have also added new text to the Discussion outlining potential improvements that could improve screening efficiency, particularly more efficient delivery of DNA beads into IVTT droplets, on pg. 10.

Despite my concerns, I feel that this is an impressive and important contribution. I recommend that this work could be acceptable for publication once the major concerns listed above, and additional concerns listed below, have been addressed.

Additional concerns to be addressed.

- The procedure requires four distinct microfluidic steps, the first two of which could presumably be combined into one step if the biotinylated primers were pre-immobilization to the beads. The authors state that pre-immobilization would limit the number of DNA copies per bead, but it is not clear from the data provided in this manuscript that this is truly a limiting factor or an important consideration. While the authors mention that pre-immobilized primers could limit amplification efficiency due to crowding, couldn't the same argument be applied to the efficiency of transcription of more densely modified beads? There must be some trade-off here, and the authors have not convinced this reader that they have explored these parameters thoroughly enough to draw strong conclusions about the optimal approach.

We appreciate the reviewer's point, and we shared similar reasoning at the outset of this project. After extensively testing amplification of ~1–2 kb templates from pre-immobilized primers on beads isolated in PCR droplets, we failed to get more than ~0.1–1% of DNA beads capable of driving strong expression of our target sensors in IVTT droplets. Briefly, we tested (in combination): 1) multiple high-performance hot-start PCR reagents, 2) multiple sizes of polystyrene beads, silica beads and magnetic Dynabeads, 3) the composition of the immobilized 3'-primers (single versus dual biotin linkages, with variable spacers between the linkage and the primer, and variable annealing regions), 4) the ratio of immobilized:free 3'-primer, 5) the size of the emPCR droplets, and of course 6) the thermocycling parameters. While we did not pinpoint the exact reasons for the poor results, ultimately, it was post-amplification bead-immobilization that reliably produced beads yielding

uniformly high expression levels of our sensors of interest. Indeed, we have found the consistency of this method to be both remarkable and necessary.

We also made other attempts to eliminate steps from library GSB production, namely 1) direct merging of emPCR droplets with IVTT droplets, and 2) expression of sensor protein in IVTT droplets pre-loaded with the agarose and alginate gels. We also investigated other means of DNA delivery into the IVTT droplets, specifically 1) DNA trapped in agarose hydrogels, and 2) purified DNA nanoparticles produced as described in Galinis et al 2016. We found that these methods were not nearly as successful as our final pipeline, both in terms of sensor expression level (highly variable, with many IVTT droplets falling below our detection limit) and the reliability of GSB production. In our revision, we have further clarified our reasons for building the pipeline as we have in the main text on pgs. 3–4.

- On page 30, it is stated that, "Biosensors were also subcloned into bacterial expression vectors so the biosensor protein could be purified and subjected to in vitro specificity testing." However, further experimental methods of bacterial expression, purification, and methods for characterization of purified proteins, seem to be missing. Furthermore, it is not exactly clear which in vitro characterization experiments were done with purified proteins, and which were done with IVTT-produced LiLac in GSBs.

We have clarified in the text that all GSB experiments with LiLac were performed with IVTT-produced LiLac in GSBs, while the photophysical characterization newly described here in our response was performed with purified LiLac protein. We have updated the Methods section to clearly describe the expression/purification methods used in this manuscript.

- The methods explains that beads were limited to ~100,000 copies of DNA per bead by pre-blocking with biotin. However, in the main text, it is stated that beads have >200,000 copies.

We have clarified in the main text on pg. 4 that beads can accommodate up to >200,000 copies, but are intentionally limited to ~100,000 copies for optimal soluble sensor expression.

- I feel that it would be more appropriate to put actual measured lifetimes on the pseudocoloured bar in Figure 5, rather than "High" and "Low".

We have updated the pseudocolor bars as the reviewer has suggested in Figures 4 and 5.

- The abstract mentions, "specificity" as a feature that can be evaluated using BeadScan, but that was not demonstrated in this work.

In this manuscript, we isolated LiLac by performing a BeadScan screen that co-varied the concentration of two ligands: lactate and H⁺ (indicated as a pH value). While we appreciate that "specificity" is often used to indicate a strong preference for one ligand over other small molecule ligands, ions are also important ligands. Indeed, the structurally-similar TlpB receptor directs pH-taxis via a pH sensing mechanism in the distal PAS domain of its extracellular binding domain (Sweeney, et al 2012). We do appreciate the reviewer's point that we did not screen for responses to lactate over other small molecules, like pyruvate or oxaloacetate, because these parameters had previously been established for the TlpC binding domain in Machuca, et al 2017. But we would respectfully argue that establishing a strong preference for lactate over H⁺ is indeed a type of specificity screen. To clarify this point, we have qualified the statement referring to "specificity" in

the main text, and expounded further in the Discussion on pg. 9 that, while we did not covary two small molecule ligands during LiLac optimization, BeadScan can also support that kind of specificity screen.

- In the Discussion, statements about the range of conditions that can be evaluated (e.g., "can be assayed against many conditions") would be toned down a bit and it should be made clear that BeadScan could "potentially" enable this.

We have changed the phrase "many conditions" to "multiple conditions" to reflect that there is a practical limitation on the number of conditions that can be screened. Here, we have performed screens with 5–8 distinct user-defined conditions, using the minimum number of conditions required to acquire the desired dataset. However, the number of conditions applied in a screen is limited primarily by the number of times the GSBs can be imaged before photobleaching effects become problematic. This is dependent on multiple factors, including the brightness of the GSBs and the photostability of the sensors within the library. While it is therefore prudent to limit the number of images collected (e.g. the number of conditions tested), bright, stable, high-expressing sensor libraries could feasibly be screened against more conditions than used here.

Reviewer #3 (Remarks to the Author):

Koveal and co-authors developed BeadScan, which is a screening method for developing fluorescent biosensors using droplet microfluidics. They first created a DNA library that codes the protein biosensor. The droplet-isolated template DNA was amplified by PCR inside the droplet (with each droplet containing 0 or 1 DNA molecule). Then the droplet was fused with another droplet containing a bead. After breaking the droplet and washing out excessive DNA, the single DNA-containing bead was re-encapsulated in a droplet that contains purified IVTT (in vitro coupled transcription/translation). Eventually, after the expression of biosensor protein, the droplet transformed into gel-shell beads (GSB) which allows the passing of small molecules while trapping the protein biosensor inside. The GSBs adhered to the glass surface and substrate for the biosensor can be added to examine the biosensor performance in terms of fluorescence lifetime. Using the BeadScan method, the authors developed a biosensor (LiLac) that could quantitatively measure lactate concentrations in living cells and tissue. The authors demonstrated the advantage of LiLac by comparing it with the existing methods Laconic in terms of response size, variability, and signal-to-noise ratio. The work in the paper was well executed. However, the level of innovation is fairly low. The droplet microfluidic operation (droplet formation and fusion) is very routine. GSB approach is interesting but was developed in 2014 by a different lab for use in biocatalyst screening (see ref 17 in the manuscript). The application to biosensor screening is a fairly trivial change from the original 2014 work.

While droplet formation and fusion technologies are generally well-established, we would propose that the innovativeness of our work should be judged by the unique nature of the new advances enabled by our system (described at length throughout the manuscript, most prominently in the Discussion) and the high degree of technical complexity and optimization that had to come together to create BeadScan. While the GSB formulation (agarose core + alginate/PAH shell) was previously described, we had to establish a completely new pipeline to make GSBs compatible for fluorescent biosensor screening. This advance was far from trivial and required labor-intensive testing of many reagents and materials used for sophisticated biochemical reactions and physical manipulations that had to be systematically optimized in concert. Furthermore, the nature of the polymers that are

handled in this system (viscous, temperature-sensitive) made it quite challenging to establish consistent microfluidic manipulation, requiring fastidious fine-tuning of our microfluidic devices. To better convey these points, we have further emphasized these advances in the main text of our revised manuscript.

Specific comments:

1. On page 3, why use the microbead to capture amplified DNA when the authors can infuse the droplet directly with IVTT system? Was there any interference between the reagents and IVTT system?

We have found the purified IVTT systems (PURExpress and PUREflex2.0) to be highly sensitive to any departure from the ideal reaction conditions, particularly dilution of the reagents. These sensitivities are also documented in the corresponding user manuals. In our hands, merging emPCR droplets with IVTT droplets results in significant dilution of the IVTT reagents and failed to yield workable sensor expression levels. Therefore, to eliminate the problem of reagent dilution, we chose to purify bead-immobilized DNA and deliver the DNA beads into IVTT droplets in a minimal volume, such that the final IVTT + DNA bead droplets have a 1X final concentration of IVTT reagents. We have included this reasoning in the main text on pg. 3 to clarify our motivation for using DNA beads.

2. On page 5, the authors need to clarify whether the lactate concentration and pH were changed in the same experiment for TlpC-TQ. If the tests of the two parameters were done sequentially, the authors need to state that.

4. An important selling point of the manuscript seems to be on multiparameter screening. However, the screening against individual parameters seems to be done separately in their experiments. These tests were done sequentially (instead of simultaneously). Thus the claim on multiparameter screening is misleading.

Because comments 2 and 4 are related, our response will address both together. First, we will clarify here how we performed the TlpC-TQ library screening using BeadScan. Concentrations of lactate and H^+ (reported as pH) were co-varied in the TlpC-TQ screen, with conditions delivered in a randomized order, as shown in Supplementary Figure 4. Specifically, a single coverslip of immobilized GSBs was exposed to the first condition (0 lactate, pH 6.7) and imaged, then exposed to the second condition (0 lactate, pH 7.6) and imaged again, and so on. So even though each individual condition was delivered sequentially, the two ligands (lactate and H^+) were physically co-varied during the screen, giving rise to the two-dimensional plot of lactate- versus pH-induced changes in lifetime shown in Figure 3G. All conditions were therefore tested in a single round of screening, on the exact same set of sensor variants.

To select the optimal sensor variant, the parameter highlighted in Figure 3G (specificity for lactate over H^+) was co-evaluated with three additional parameters: 1) affinity for lactate, 2) response size and 3) intensity. This is, in fact, a routine application of BeadScan. Because we are able to extract all of these parameters, or "features," from a set of 5–8+ images collected under distinct conditions and use them simultaneously to evaluate sensor variant fitness, we would argue that BeadScan qualifies as a multiparameter screen.

3. On page 5, since mTurquoise2 was used to develop the fluorescent lifetime sensor for calcium level, and the authors mentioned on page 2 that one of the limitations for existing lactate biosensor was an

undesirable response to calcium. The authors should test LiLac in response to the change of calcium level.

*The strong calcium response of one previously developed lactate biosensor, eLACCO1, almost certainly derives not from the fluorescent protein, but from the scaffold periplasmic binding protein that was used, which binds a complex of lactate and calcium ion (as shown in crystal structures of both the PBP and the eLACCO1 sensor protein). Nevertheless, we agree that it is critical to establish calcium-independence for our lactate sensor, and we respectfully direct the reviewer to Figure 4A, which shows that LiLac does not show a response to calcium at very high concentrations (1 mM). We have further highlighted this point in the main text on pg. 6, with the following bolded addition: LiLac is highly specific for lactate over other ligands, including pyruvate **and calcium** (Fig 4A).*

REVIEWERS' COMMENTS

Reviewer #1 (Remarks to the Author):

The authors have addressed my minor concerns.
I deem this manuscript suitable for publication in this journal.

Reviewer #2 (Remarks to the Author):

The authors have thoroughly and thoughtfully addressed all of the reviewer comments. I am satisfied with the revisions and recommend that the manuscript be accepted in its current form.

Reviewer #3 (Remarks to the Author):

The revised manuscript has added clarifications on the work's novelty and justification for some technical steps. The revision is much improved. I am fine with publication of the manuscript.